# Bioactive Compounds Formulated in Phytosomes Administered as Complementary Therapy for Metabolic Disorders

**DOI:** 10.3390/ijms25084162

**Published:** 2024-04-09

**Authors:** Laura Toma, Mariana Deleanu, Gabriela Maria Sanda, Teodora Barbălată, Loredan Ştefan Niculescu, Anca Volumnia Sima, Camelia Sorina Stancu

**Affiliations:** Institute of Cellular Biology and Pathology “Nicolae Simionescu” of the Romanian Academy, 8 B.P. Haşdeu Street, 050568 Bucharest, Romania; laura.toma@icbp.ro (L.T.); mariana.deleanu@icbp.ro (M.D.); gabriela.sanda@icbp.ro (G.M.S.); teodora.barbalata@icbp.ro (T.B.); loredan.niculescu@icbp.ro (L.Ş.N.); anca.sima@icbp.ro (A.V.S.)

**Keywords:** bioactive compounds, cardiovascular diseases, diabetes mellitus, dyslipidemia, hepatic disorders, inflammatory stress, metabolic disorders, metabolic syndrome, oxidative stress, phytosomes

## Abstract

Metabolic disorders (MDs), including dyslipidemia, non-alcoholic fatty liver disease, diabetes mellitus, obesity and cardiovascular diseases are a significant threat to human health, despite the many therapies developed for their treatment. Different classes of bioactive compounds, such as polyphenols, flavonoids, alkaloids, and triterpenes have shown therapeutic potential in ameliorating various disorders. Most of these compounds present low bioavailability when administered orally, being rapidly metabolized in the digestive tract and liver which makes their metabolites less effective. Moreover, some of the bioactive compounds cannot fully exert their beneficial properties due to the low solubility and complex chemical structure which impede the passive diffusion through the intestinal cell membranes. To overcome these limitations, an innovative delivery system of phytosomes was developed. This review aims to highlight the scientific evidence proving the enhanced therapeutic benefits of the bioactive compounds formulated in phytosomes compared to the free compounds. The existing knowledge concerning the phytosomes’ preparation, their characterization and bioavailability as well as the commercially available phytosomes with therapeutic potential to alleviate MDs are concisely depicted. This review brings arguments to encourage the use of phytosome formulation to diminish risk factors inducing MDs, or to treat the already installed diseases as complementary therapy to allopathic medication.

## 1. Introduction

Metabolic disorders (MDs) are the main cause of life-threatening diseases such as diabetes mellitus (DM), cardiovascular diseases (CVDs) and liver pathologies, affecting people worldwide, despite the various therapies developed for their treatment [1,2,3,4,5,6]. Connections between these pathologies were established, and the underlying mechanisms interconnecting them are intricated and involve the activation of different molecular pathways [7,8,9]. Extensive experimental and clinical evidence identified oxidative stress and inflammatory stress as common denominators and important players in the inception and progression of MDs [7,8,10]. Risk factors such as hyperglycemia, increased levels of advanced glycation end-products and of free fatty acids, decreased levels of high-density lipoprotein cholesterol (HDL-C) that appear in dyslipidemia, obesity, DM or non-alcoholic fatty liver disease (NAFLD) determine the activation of pro-oxidative nicotinamide adenine dinucleotide phosphate (NADPH) oxidase (NADPHox), the uncoupling of endothelial nitric oxide synthase (eNOS) or the dysfunction of mitochondria. Thus, the increased production of oxygen reactive species (ROS) is determined. In parallel, the downregulation of the antioxidant defense system including superoxide dismutase (SOD), catalase (CAT) and the enzymes involved in the metabolism of glutathione (GSH) is induced, resulting in a diminished detoxification of ROS [8,10]. The increased oxidative stress further acts as a trigger for the inflammatory stress, determining the increase in tumor necrosis factor alpha (TNF α), C-reactive protein (CRP), nucleotide-binding oligomerization domain (NOD)-like receptor protein 3 (NLRP3) inflammasome, nuclear factor kappa B (NF-kB), mitogen-activated protein kinases (MAPK), Janus kinases (JNKs), interleukins (ILs), etc., further exacerbating the pathological processes [8]. In addition, the excessive levels of ROS contribute to the oxidation of low-density lipoproteins (LDLs) and HDLs, endothelial cell dysfunction or apoptosis, vascular remodeling and atheroma formation [11], contributing to the development of cardiovascular diseases (CVDs), one of the major complications of NAFLD, obesity or diabetes.

Many therapies designed to treat MDs have failed completely or partially (cholesteryl ester transfer protein inhibitors, troglitazone, vitamins), or were too expensive to be applied to the entire population at CVD risk (apolipoprotein A-I Milano). Other treatments have been successful, but they induce considerable side effects (statins, sulfonylureas, calcium channel blockers, renin–angiotensin system inhibitors) [12,13,14]. Therefore, the exploration for new products to prevent or treat MDs is of high and continued interest. In the last decade, scientific researchers turned their attention to phytochemicals as effective, safe and low-cost natural bioactive compounds for MD treatment [15,16,17,18]. Phytochemicals such as polyphenols, alkaloids and terpenoids have good antioxidative activity and anti-inflammatory effects, due to their ability to bind free radicals with their functional groups. The main signaling pathways by which numerous natural bioactive compounds exert anti-oxidant and anti-inflammatory actions involve the modulation of transcription factors, the inhibition of NLRP3 inflammasome and NF-κB, activation of nuclear factor erythroid 2-related factor 2 (Nrf2) and protein kinase B (PKB/Akt), and the consequent stimulation of eNOS, of antioxidant enzymes and the inhibition of NADPHox [19,20,21] (Figure 1). In addition, notable lipid-regulatory effects have been described for some phytochemicals; they inhibit the lipid absorption in the small intestine, stimulating the excess cholesterol excretion via the gallbladder or small intestine, and impeding de novo lipid synthesis in the liver. The molecular mechanisms responsible for the lipid-lowering effects of some natural compounds involve the activation of essential transcription regulators, such as sirtuin 1 (SIRT-1), liver X receptors (LXRs) and the peroxisome proliferator-activated receptors (PPARs) [19,22,23,24] (Figure 1). 

In addition, it is widely accepted that drugs derived from plant extracts are safer than their synthetic equivalents [25]. It has been proven that natural compounds administered as combined total extracts exert additive effects compared to their individual counterparts’ administration [26]. Despite all these benefits acknowledged for phytochemicals, there are no substantial clinical studies to prove the clear advantages of the therapies with phytochemicals in CVDs, DM or hepatic disorders [18,27,28]. An explanation for the lack of consistent data from clinical studies could be the limited solubility and low stability of the natural bioactive compounds in circulation [18,28]. The polyphenols are compounds with very high antioxidant and anti-inflammatory potential, but their bioavailability is very limited, due to their rapid enzymatic modification in vivo that blocks the hydroxyl groups which exert their antioxidant effect [19]. Another category of compounds that cannot fully exert their beneficial properties are phytochemicals which cannot pass through the intestinal membranes by passive diffusion, due to their large chemical structure [29]. For an ideal bioavailability, the phytochemical molecules should be able to exhibit a hydrophilic–lipophilic balance [30]. Therefore, the delivery system must be developed to increase the solubility and stability of the bioactive compounds from extracts. To overcome these limitations, researchers developed new advanced drug delivery systems for phytochemicals. There are many types of nanosystems that have been created for therapeutic purposes and classified in inorganic nanoparticles (silica nanoparticles, magnetic nanoparticles, carbon nanotubes, polymeric nanoparticles, polymerosomes) and organic nanoparticles, the latter including liposomes (50–450 nm), phytosomes (50–10,000 nm), ethosomes (50–400 nm/2000–5600 nm), transferosomes (<300 nm), nanostructured lipid carriers (50–500 nm), cubosomes (100–500 nm), solid lipid nanoparticles (10–1000 nm), hexosomes, nanoparticles (10 to 1000 nm), niosomes (100–2000 nm), nanocrystals (<1000 nm), nanomicelles (10–100 nm), nanoemulsions (100–600 nm, 10–1000 nm), nanofibers (50–1000 nm) and microspheres (6–735 µm) [30,31,32,33,34,35]. 

Among these, the phyto-phospholipid complexes, named phytosomes, are self-assembling vesicular structures in aqueous environments, forming a unique and stable arrangement due to the electrostatic interactions and hydrogen bonds formed between the polar head of phospholipid (ammonium and negative phosphate groups) and the functional groups of phytochemicals having an active hydrogen atom (-COOH, -OH, -NH_2_) (Figure 2). 

There are important physico-chemical differences between phytosomes and liposomes, giving important advantages to the former: (i) the size of phytosomes is smaller than that of liposomes; (ii) the phytosomes are more stable due to the hydrogen bonds formed between phospholipids and the carried active compounds, whereas no chemical bonds are formed in liposomes; (iii) the molar ratio of phospholipids and natural compounds is 1:1 or 2:1 in phytosomes, which confers a very low susceptibility to oxidation, while in liposomes there are hundreds or even thousands of oxidation-prone phospholipids surrounding the water-soluble compounds [36]. 

The phytosomes help to minimize the loss of bioactive compounds along the digestive tract by limiting their degradation by the digestive enzymes and microbiota, improving their retention in the small intestine and increasing their absorption. Thus, the phytosomes’ formulation increases the bioavailability of phytochemicals, strengthening their efficacy, and allowing for an in vivo non-invasive delivery (oral or topic administration) [18,36]. The phytosomes containing phytochemicals may be helpful for patients who are intolerant to allopathic drugs or those who are not able to attain targeted parameters at the maximally tolerated dose of the allopathic drug. 

The rationale of the present review is to highlight the scientific proof for the improved benefic action of phytochemicals formulated in phytosomes compared to their free form in order to stimulate the use of these phyto-phospholipid complexes as remedies for MD, administered as complementary therapies to allopathic drugs. We briefly present specific methods for the preparation and characterization of phytosomes, followed by scientific evidence of their increased stability and bioavailability compared to unformulated phytochemicals. We depict the data about the use of phytosomes with bioactive compounds or plant extracts to safely alleviate oxidative and inflammatory stress, dyslipidemia, hyperglycemia and insulin resistance, the main risk factors in MD evolution.

The strategy to identify the pre-clinical and clinical studies that investigated the therapeutic potential of bioactive compounds formulated in phytosomes involved searching for publications on PubMed, Scopus and Web of Science from 2000 to 2024, with restriction to English language. The main keyword used was “phytosomes”, and the secondary keywords were “metabolic disorders”, “oxidative stress”, “inflammatory stress/inflammation”, “dyslipidemia”, “diabetes”, “metabolic syndrome”, “hepatic disorders/NAFLD” and “cardiovascular diseases”. The resulting total number of papers was 941, out of which we excluded the duplicates and retained only the articles with the full text available (excluded proceeding papers and conference abstracts). Thus, the final number of papers that met the selection criteria was 132. Using the same exclusion criteria, the number of papers that addressed the methods of phytosome preparation and characterization, and the techniques for measuring the bioavailability of active compounds from phytosomes, was 57.

## 2. Preparation and Characterization of Phytosomes with Bioactive Compounds

### 2.1. Preparation of Phytosomes with Bioactive Compounds

Phytosomes are prepared by complexing the natural bioactive compounds with phospholipids in a suitable solvent. The phospholipids are glycerophospholipids such as phosphatidylcholine (PC), phosphatidylethanolamine, phosphatidylserine, phosphatidic acid, phosphatidylinositol and phosphatidylglycerol. Among these phospholipids, PC is the most frequently used to prepare phyto-phospholipid complexes because it exhibits amphipathic properties which ensure moderate solubility in water and lipid environments [36]. In addition, PC shows low toxicity and robust biocompatibility because it is an essential component of the cell membrane. Different solvents have been used for phyto-phospholipid complex formulation. Aprotic solvents, such as methylene chloride, chloroform, tetrahydrofuran, dimethylsulfoxide, ethyl acetate or acetone have been used to prepare phytosomes, but they have been largely replaced by protic solvents like ethanol, because it is a food grade solvent, leaves less residues and causes minimal damage [37,38]. Regarding the stoichiometric ratio of bioactive compounds and phospholipids, the phyto-phospholipid complexes are employed by reacting the ingredients in a molar ratio ranging from 0.5 to 2.0 [39]. The stoichiometric ratio of 1:1 was considered optimal in many studies to prepare phyto-phospholipid complexes [30], but this ratio should be adjusted every time a new combination of phytochemicals and phospholipids is tested. The solvent, stoichiometric ratio of active ingredients, reaction temperature and reaction duration are the primary variables that influence the formation of phyto-phospholipid complexes [40]. Phytosomes can be prepared by three main techniques: solvent evaporation/thin film hydration, anti-solvent precipitation and freeze–drying [18]. A widely used technique for preparing phyto-phospholipid complexes is the solvent evaporation/thin-layer hydration method, according to which the PC and standardized extracts or bioactive compounds are combined in a round bottom flask and dissolved in a proper solvent by heating at an ideal constant temperature for a predetermined time duration. The complexes prepared in this way can be finally obtained by evaporating the solvent under vacuum. The resulting dry film can be sieved and stored in a desiccator or can be hydrated with the optimum volume of water, followed by controlled sonication in order to obtain a certain size for the phytosome particles. Then, the phyto-phospholipid complexes can be freeze–dried for long-term preservation, and later formulated in capsules (Figure 3) [38,41,42].

The anti-solvent precipitation technique is the second most used method for the preparation of phytosomes. According to this technique, the phyto-phospholipid complexes obtained as in the previously described method are finally precipitated by using a suitable solvent, such as n-hexan [43,44]. By using the freeze–drying method, the bioactive compound solubilized in the suitable solvent is mixed with the phospholipid alcoholic solution, allowed to react in predetermined conditions followed by lyophilization in order to obtain the phytosomes [42].

### 2.2. Characterization of Phytosomes with Bioactive Compounds

Several techniques were employed to establish the phytosomes’ size, composition, morphology and other physical characteristics. Some of the physical properties of phytosomes can be explored by more than one technique. The main parameters for characterization of phytosomes are: (i) size and shape; (ii) surface charge; (iii) chemical composition and structure; (iv) stability; (v) encapsulation efficiency (EE%) and (vi) release behavior. The size and morphology of particles are important indicators of the phytosomes’ quality. Different techniques can be used for phytosome size characterization, such as dynamic light scattering (DLS) [18,38,45,46,47], transmission electron microscopy (TEM) [18,38,47,48,49,50], scanning electron microscopy (SEM) [18,47,48,49,51,52], atomic force microscopy (AFM) [53] and size-exclusion chromatography (SEC) [54]. The measurement of size distribution and polydispersity of phytosomes gives information about their physical stability; it can be evaluated by DLS, an easy, very fast and precise method [18,38,45,46,47]. Zeta potential represents the complete charge generated by medium, defines the surface charge of phytosomes in emulsions and reflects the stability of the phytosomes in the medium. This parameter can be negative, positive or neutral, depending on the composition of the phytosomes. A zeta potential greater than +30 mV or less than −30 mV is known to correspond to a stable emulsion of phytosomes [51,52,55,56]. The electrostatic properties of phytosomes can be measured using Zeta Sizer [51,52,55,56]. The chemical composition and interaction between phospholipid and phytochemicals are evaluated by high-performance liquid chromatography (HPLC) [38,46,52] and liquid chromatography coupled to mass-spectrometry (LC-MS) [51]. To characterize the solid-state matter in the complex form, the thermal analysis is widely used. In differential scanning calorimetry (DSC), the determination of changes in solid-state properties, in accordance with the temperature change, provide important information about the phytosomes’ stability, degradation and melting (Table 1). Phyto-phospholipid complexation and molecular interactions in solution are studied by employing spectroscopic techniques like proton (^1^H) nuclear magnetic resonance (NMR), carbon-13 (^13^C) NMR, phosphorus-31(^31^P) NMR, and Fourier transform infrared (IR) spectroscopy (FTIR). Formation of the hydrogen bonds is associated with specific signals like changes in chemical shift and line broadening in NMR spectra and with appearance of new bands in IR spectra [38]. Studies of the phytosomes’ stability are performed to explore the phytochemical changes in phytosomes during storage, and can be measured over several months by determining average size, zeta potential, size distribution and drug content with the same techniques. Determination of the encapsulation efficiency (EE%) begins with the removal of free, unencapsulated phytochemicals from the phytosomes emulsion by different methods, such as ultracentrifugation [18,38,46,52,57,58], the Sephadex gel column or dialysis method (specific cut-off) for several hours against buffer solution [18,45]. The formula for EE% calculation is as follows:EE (%) = [(Initial amount of phytochemical − amount of free phytochemical from filtrate)/initial amount phytochemical] × 100

The amounts of free (unloaded) phytochemicals in filtrate could be measured spectrophotometrically (UV–Vis) [47,58] or by HPLC, ultra-performance liquid chromatography (UPLC) or LC-MS methods [38,46,49,52]. The release of phytochemicals from phytosomes can be abrupt or sustained. In vitro drug release is explored by introducing the phytosomal solution into a dialysis membrane which is dipped into a container containing release medium at a constant stirring speed and temperature. The phytochemical released in the medium is quantified by spectrophotometry or HPLC, LC-MS methods. The established and accepted methods for the characterization of phytosomes with bioactive compounds are summarized in Table 1.

## 3. Evaluation of the Bioavailability of Active Compounds Formulated in Phytosomes

The phyto-phospholipid complexes being amphiphilic allows for a better dissolution in the gastrointestinal fluid and very good absorption through the lipophilic membranes or intestinal cells. Moreover, Wang H. et al. [66] indicated that the phospholipid molecules have the potential to act as chaperones for the drugs, transporting them through the biological membranes and improving their bioavailability. They also improve the drugs’ stability by encapsulating them into the formed nanocarriers and protecting the drug molecules from the aggressive environment in the gastrointestinal tract. Thus, the phyto-phospholipid complexes can be transported via the microfold (M)-cell of the intestinal mucosa [28]. Many studies have been dedicated to the development and in vitro characterization of phytosomes, but in vivo studies need to be conducted to prove their effectiveness and to determine their oral bioavailability that is expressed as the rate and extent of the bioactive compound absorption which can reach into the bloodstream. The experiments for measuring the bioavailability of a bioactive compound can be achieved in animal models or humans. After in vivo administration of the phytosomes and unformulated extracts, the amount of drug entering into the systemic circulation is measured at different time points and plotted to acquire the maximum drug concentration (Cmax), peak time (Tmax) and area under the curve (AUC). These are the kinetic parameters that can be used to characterize the properties of phytosomes in terms of bioavailability [67]. The relative bioavailability of a bioactive compound can be calculated by dividing the value measured for AUC of the compound in the phytosome by the AUC value of the unformulated compound. The main parameters to be measured in order to establish the level of bioavailability of the bioactive compounds formulated in phytosomes compared to unformulated compounds are summarized in Table 2. An important advantage of the phytosomes is that they can be administered through non-invasive routes. Thus, most of the products based on phytosomes are designed for oral administration (see Table 3), but there are also phytosomes created for nasal [68] or topic administration [69].

The commercially available products based on bioactive compounds or plant extracts formulated into phytosomes and suitable for prevention or treatment of MD are reviewed in Table 3.

## 4. Pathologies Addressed by Natural Bioactive Compounds Formulated in Phytosomes 

### 4.1. Oxidative and Inflammatory Stress

It is generally accepted that oxidative and inflammatory stress are responsible for the initiation and progression of numerous diseases, including MD, and many therapies have been attempted, but without remarkable results [27]. In the last quarter century, numerous phytochemical formulations in phytosomes have been prepared and demonstrated to have antioxidant and anti-inflammatory effects. Here, we display those with potential impact on MDs. The first indication that a chemical bond is formed between phospholipids and natural bioactive compounds in phytosomes came from the group of Bombardelli E. in 1989 [94]. After this moment, an increasing number of laboratories have approached and developed this biotechnology. 

#### 4.1.1. Preclinical Studies

The extracts of *Ginkgo biloba* leaves have been found to possess multiple beneficial properties, the main active constituents being flavone glycosides, such as kaempferol, quercetin and isorhamnetin [95]. The Ginkgoselect Phytosome^®^ (Indena SpA, Milan, Italy) prepared by mixing a stoichiometric amount of soy phospholipids and *Ginkgo biloba* leaf extract has been tested for antioxidant properties in Wistar rats with rifampicin-induced hepatotoxicity (500 mg/kg, for 30 days). Simultaneously administered, Ginkgoselect Phytosome^®^ at 25 mg/kg and 50 mg/kg significantly lowered the plasma levels of lipid peroxides, and elevated the amounts of GSH, SOD, CAT, glutathione peroxidase (GPx) and glutathione reductase (GR) in liver homogenates, in a dose-dependent manner. No significant differences between the groups treated with phytosomes or silymarin (100 mg/kg, positive control) were observed [96].

The bioactive compound most frequently formulated in phytosomes and tested is curcumin, the primary polyphenol found in turmeric (*Curcuma longa*) [18]. The antioxidant activity of curcumin is due to its ability to scavenge radicals generated in oxidation processes, and its anti-inflammatory effects are based on the down-regulation of cyclooxygenase-2, MAPK and JNK, and the prevention of TNFα and ILs production [97,98]. Curcumin formulated in nanophytosome (at a dose of 15 mg/kg for seven days) improved the antioxidant effects in a mouse model of inflammation [99]. Acute inflammation in the mice was induced by the administration of carrageenan (1%) into the subplantar region of the paw. The pre-treatment with curcumin formulated in nanophytosomes induced a significant increased enzymatic activity of CAT, SOD, GR and GPx, compared to unformulated curcumin [99]. This study brings evidence that the phytosomal formulation of curcumin improves its therapeutic potential.

Ginger (*Zingiber officinale*) rhizomes are used as food and drink spices and in traditional medicine for their antioxidant, anti-inflammatory, hypolipidemic, antidiabetic and anticoagulant effects [100,101,102,103,104]. The most abundant bioactive compounds of the ginger extract are gingerols and shogaols, able to trigger multiple signaling pathways to exert their therapeutic effects, but having low bioavailability due to their low solubility [105]. Therefore, high doses of ginger extracts are used with the intent to increase its benefits. Some side effects have been described in these conditions, such as digestive tract irritation, bleeding or cardiac arrhythmias [106]. The rosehip fruits (*Rosa canina*) have been used in traditional medicine to reduce pain or as hypolipemic and hypoglycemic treatment, being characterized as having antioxidant, anti-inflammatory, anti-obesity, hepatoprotective, nephroprotective and cardioprotective properties [107,108]. The major bioactive constituents of rosehips, flavonoids, anthocyanins and phenolic compounds are lipophilic and present low bioavailability. Thus, Deleanu M. et al. [38] developed a new formulation of phytosomes with bioactive compounds from ginger and rosehips extracts, and demonstrated that those with the mass ratio ginger extract:rosehip extract:phosphatidylcholine—0.5:0.5:1 had the most effective antioxidant and anti-inflammatory effects in cultured human enterocytes. Moreover, they showed that the phytosome formulation had doubled the plasma concentration of bioactive compounds, compared to unformulated extracts, by increasing their absorption from the digestive tract and accumulation in the liver and kidneys of C57Bl/6J mice. In addition, Deleanu et al. [38] evidenced that the antioxidant and anti-inflammatory effects of the extracts were significantly improved by the phytosome formulation. These properties have been demonstrated in mice with lipopolysaccharide (LPS)-induced systemic inflammation as the increased plasma level of SOD2 and paraoxonase 1 (PON1), and decreased protein expression of TNFα and IL-1 beta (IL-1β) in the liver and small intestine. All these data confirm the enhanced therapeutic properties of ginger and rosehip extracts formulated in phytosomes.

Gallic acid is a well-known flavonoid and the main bioactive compound of sumac (*Rhus coriaria* L.), having antioxidant, anti-inflammatory and antitumoral activity [109,110]. Despite these beneficial activities, the therapeutic potential of gallic acid is limited due to its low absorption, poor bioavailability and rapid elimination [111,112]. The group of Abbasalipour H. et al. [113] formulated gallic acid and sumac extract in phytosomes and tested their antioxidant properties at doses of 20 mg/kg daily in a rat model of valproic acid-induced oxidative stress in the nervous system. Both types of phytosomes showed superior antioxidant effects measured as increased plasma activity of GPx, GR, SOD, CAT and GSH level, compared to unformulated gallic acid or sumac extract [113]. The mechanism involves the Nrf2 signaling pathway that regulates the expression of various antioxidant enzymes and is activated as a defense mechanism against oxidative stress [114]. Gallic acid and sumac extract formulated in phytosomes, compared to the unformulated extract, manifested better solubility and bioavailability, induced the doubling of Nrf2 gene expression and blocked the binding of Nrf2 to Kelch-like ECH-associated protein 1 (Keap1), leading to the activation of downstream antioxidative enzymes [113]. These results show that the phytosomal formulation of plant extracts or of their active constituents can improve their beneficial effects.

Berberine is a benzylisoquinoline alkaloid found in many plants, particularly in barberry (*Berberis vulgaris* L. and *Berberis aristata* DC.), and has antihypertensive, hypoglycemic and hepatoprotective effects [115,116,117]. It has high water solubility, but its bioavailability is low when orally administered due to its high molecular weight and self-aggregation tendency that impede its passage through the intestinal wall. Thus, the absolute bioavailability of berberine is low, and high doses are necessary for administration in clinical studies, which unfortunately causes adverse gastrointestinal effects [52,118]. To increase berberine bioavailability and efficacy, the group of Güngör-Ak A. et al. [29] used the Quality by Design method for the formulation design and selection of the optimum formula to prepare berberine–phospholipid complexes using the reverse phase evaporation method. The resulting berberine phytosomes have a small particle size and narrow particle size distribution. The antipyretic activity was detected in rats only at high dose (209 mg/kg versus 104.5 mg/kg), and for longer periods of time (from 6–10 days to 8–14 days) compared to unformulated berberine. The analgesic and anti-inflammatory effects of berberine in phytosomes have been shown for the high dose only (209 mg/kg), which reduced the hind paw edema induced by carrageenan and serotonin, the subcutaneous air-pouch Freund’s complete adjuvant-induced inflammation, and inhibited the acetic acid-induced capillary permeability in mice [29].

The wild leek (*Allium ampeloprasum*) is known for its nutritional and therapeutic properties as antiplatelet, antidiabetic and anti-atherosclerotic effect [119]. Very recently, the group of Shoeibi A. et al. [120] formulated in phytosomes a fraction enriched in polyphenols extracted from leek and tested their antioxidant properties in BALB/c mice with colon carcinoma. Administered at a dose of 50 mg total phenolic compounds/kg for 28 days, the phytosomes increased the gene expression of GPx and SOD in the liver, and reduced the lipid peroxides measured as malondialdehyde (MDA) levels, compared to the group receiving unformulated leek extract [120]. 

The *Gymnema inodorum* leaf extract contains bioactive compounds, such as phenolic acids, flavonoids, triterpenoids and pregnane glycosides, which have been shown to have antioxidant and anti-inflammatory properties and are beneficial for diabetic control, but have low bioavailability [121,122]. Very recently, Nuchuchua O. et al. [26] developed phytosomes with *G. inodorum* extract to overcome this limitation. The formulation of *G. inodorum* phytochemicals in phytosomes significantly changed the particles’ surface charge from neutral (blank phytosome) to negative (−35 mV to −45 mV), allowing for the bioactive compounds to be embedded in the phospholipid membrane. Nanoparticles at sizes lower than 200 nm show direct diffusion via the intestinal mucosal sites. Phytosomes with *G. inodorum* extract exhibited increased anti-inflammatory activity in LPS-stimulated RAW 264.7 macrophages, by lowering the production of nitric oxide (NO) induced by pro-inflammatory conditions, compared to unformulated *G. inodorum* extract [26]. All these data confirm the enhanced antioxidant and anti-inflammatory effects of plant extracts following the phytosomal formulation.

Tripterine, also known as celastrol, is a phytochemical derived from the plant of *Trypterygium wilfordii* and belongs to the family of quinone methides [123]. Tripterine exhibits anti-inflammatory effects by modulating proinflammatory cytokines, such as IL-18 and IL-1β, and inhibiting the effects of LPS or interferon gamma (IFNγ) in macrophages [124]. Due to its low bioavailability and certain toxicity at high doses, tripterine has limited clinical applications; thus, phytosome formulation was used as a method to increase tripterine absorption and sustained release. In addition, selenium supplementation of phytosomes for the improvement of SOD was also used to increase the beneficial effects of tripterine [125,126,127]. Pyroptosis is an inflammasome-mediated programmed cell death dependent on caspase-1, and it may be a major cause of multiple organ dysfunction [128]. Thus, in the attempt to develop new drugs to treat inflammatory diseases, Liu S. et al. formulated tripterine into phytosomes and functionalized them with selenium (Se@Tri-PTs) to attenuate the cytotoxic effect of phytochemical and potentiate its anti-inflammatory effect [124]. Formulation in phytosomes increased the tripterine solubility by ~500-fold and their uptake by murine J774A.1 macrophage, while the cytotoxicity decreased. Se@Tri-PTs inhibited the activation of NRLP3 inflammasome and pyroptosis by reducing the cleavage of gasdermin D and release of IL-1β in a dose-dependent manner (in a rage of 50–200 ng/mL) [124]. Thus, tripterine becomes safer and effective following formulation in phytosomes.

For hesperidin, a flavanoid within the flavanone subclass, numerous biological effects have been described, such as anti-inflammatory, heart and blood vessels protection, anti-diabetic and neuroprotective [19,129,130]. Due to its poor water solubility, which is further diminished by the acidic environment, the reported bioavailability is low (<25%) and the transmembrane permeability is reduced following oral administration. The group of Kalita B. and Patwary B.N. [131] formulated phospholipid complexes with hesperidin and demonstrated their enhanced solubility in a basic buffer system and a very balanced partition coefficient, suggesting a good membrane permeation efficiency. In addition, the complexes showed a concentration-dependent increase in the anti-oxidant activity, similar to ascorbic acid [131].

#### 4.1.2. Clinical Studies

In the late nineties, Nuttall S.L. et al. [132] prepared an extract from grape seeds formulated in phytosomes (Leucoselect^TM^ Phytosome^®^, Indena SpA, Milan, Italy), starting from the observation that, in countries with increased consumption of red wine, the incidence of coronary artery disease is lower compared to other countries. Grape seed proanthocyanidins have been reported to inhibit lipid peroxidation, capillary fragility and platelet aggregation, and regulate the activity of phospholipase A2, cyclooxygenase and lipoxygenase [133,134]. Nuttall S.L. et al. [132] administered an equivalent of 300 mg of grape procyanidins to 20 young healthy volunteers and induced a doubling of the serum total antioxidant activity for over 3 h from ingestion. This observation marked a change in perception according to which antioxidant consumption is part of a healthy diet for the concept of therapeutic strategy in diseases which are known to be aggravated by oxidative stress. 

Studies have shown that, when administered orally, curcumin has a very short half-life (10 min) due to its rapid metabolization, low absorption in the gastro-intestinal tract and rapid excretion [135,136,137,138,139]. In order to boost curcumin activity, its complexation with phospholipids to form phytosomes seems to be the most promising technology, due to the demonstrated improvement in its intestinal absorption and metabolic stability [139]. Recently, it has been shown that a six months treatment with curcumin formulated in phytosomes (Meriva^®^, Indena SpA, Milan, Italy, 500 mg tablet × 2/day, equivalent of 100 mg curcuminoids/tablet) significantly reduced plasma pro-inflammatory mediators, such as monocyte chemoattractant protein-1 (MCP-1/CCL-2), IFNγ and IL-4, as well as lipid peroxidation products (thiobarbituric acid reactive substances, TBARS) in chronic kidney disease patients [140]. Gut microbiota has been also influenced by Meriva^®^ treatment, the *Escherichia-Shigella* being significantly lowered, while *Lachnoclostridium* and *Lactobacillaceae* spp. being considerably increased, the latter being known as playing an important role in the maintenance of the gut barrier function. It is important to mention that no adverse effects have been observed in the treatment group, confirming the good tolerance of curcumin phytosomes even on long-term administration [140]. 

### 4.2. Dyslipidemia

Dyslipidemia, defined as increased plasma concentrations of total cholesterol (TC), low-density lipoproteins cholesterol (LDL-C), triglycerides (TGs), low plasma concentrations of HDL-C or a combination of these, is a well-known major risk factor for CVD [141]. Dyslipidemia can trigger the accumulation of lipids in the arterial wall, which results in the development of the atherosclerotic plaque, the underlying cause of CVD [142]. Many types of drugs have been developed in order to decrease circulating levels of cholesterol. Among these, statins are one of the most widely used. They target endogen cholesterol synthesis, by inhibiting hydroxymethylglutaryl coenzyme A (HMG-CoA) reductase, the rate-limiting enzyme in the cholesterol biosynthesis pathway. Another class of drugs used for this purpose are Niemann-Pick C1-Like 1 (NPC1L1) inhibitors, such as ezetimibe, which inhibits the absorption of cholesterol in the intestine [143]. A novel class of lipid-lowering drugs that has emerged are monoclonal antibodies, such as evolocumab, an inhibitor for proprotein convertase subtilisin/kexin type 9 (PSCK9), a known ligand of LDL receptor (LDLR) [144]. One common aspect for all these drugs are their side effects, such as dizziness, nausea, muscle weakness or headaches [145]. This is why the complementary therapies to all these drugs are the bioactive compounds that present lipid-lowering properties with minimal side effects. Therefore, their formulation in phytosomes may be a valid complementary therapy to allopathic lipid-lowering drugs. 

#### 4.2.1. Preclinical Studies

The curry tree (*Murraya koenigii*) is a rich source of alkaloids, coumarins and phytochemicals such as girinimbin, iso-mahanimbin and koenimbine, which are known to have antioxidant, antidiabetic and lipid-lowering properties, but have low bioavailability due to their limited absorption through biological membranes [146]. Therefore, the group of Rani A. et al. [51] prepared phytosomes containing *M. koenigii* extract to improve its bioavailability and assessed the hypolipidemic effects in vivo, using streptozotocin (STZ)-induced diabetic Wistar rats. The animals were treated for 21 days with phytosomes (100 mg/kg and 200 mg/kg) or unformulated *M. koenigii* extract (200 mg/kg and 400 mg/kg). Animals displayed significantly reduced levels of serum TC, TG, LDL-C and very low-density lipoproteins cholesterol (VLDL-C), and increased levels of HDL-C compared to the untreated group. The phytosomes-treated group showed a tendency for lower concentrations of TC, TG, LDL-C and VLDL-C and for higher concentrations of HDL-C compared to the group treated with unformulated extract, but these differences were not statistically significant [51]. 

*Crataegus aronia* is recognized to have lipid-lowering and antidiabetic properties [147,148]. The ethanolic leaf extract was found to be strongly enriched in phenols, flavonoids, alkaloids and tannins. Altiti A.J. et al. [50] formulated the *Crataegus aronia* leaf ethanolic extract in phytosomes and evaluated their potential to improve the lipidic profile of STZ-induced diabetic Wistar albino rats, by comparing the effects to those of unformulated extract at similar concentrations (150 and 250 mg/kg). Both the phytosomes and the extract displayed lipid-lowering properties, by reducing the serum levels of TC, TG and LDL-C and by increasing the levels of HDL-C at 21 days after the administration, but there was no statistically significant difference between the two groups in this respect [50]. These results show that, in some cases, the phytosomal formulation may not improve the therapeutic effects of the bioactive compounds.

Caffeic acid is a very active hydroxycinnamic acid, being found in fruits, vegetables, coffee and wine [149]. It possesses many health benefits, among which is the lipid regulatory effect, but caffeic acid displays poor water solubility and poor oral absorption [150]. To improve these aspects, Mangrulkar S. et al. [151] formulated caffeic acid in phytosomes by encapsulating the bioactive compound in Phospholipon^®^ 90H (Lipoid, Ludwigshafen, Germany). They evaluated the lipid regulatory potential of the phytosomes on Sprague Dawley rats fed a high fat diet for 8 weeks. The results show that formulated caffeic acid significantly lowered the levels of TC, TG, LDL-C, VLDL-C and increased the level of HDL-C, proving superior lipid regulatory action compared to the pure caffeic acid, at the same concentration of 40 mg/kg [151].

The effects of curcumin formulated in phytosomes on atherosclerosis induced by a high fat diet in New Zealand white rabbits were investigated by the group of Hatamipour M. et al. [152]. Curcumin phytosomes (Meriserin, Indena SpA, Milan, Italy) at two doses equivalent to 10 and 100 mg/kg of curcuminoids were administrated for 4 weeks. The results showed a significant reduction in the atherosclerotic plaque histopathologically evaluated on sections of the aortic arch. In addition, the intima/media thickness ratio and the macrophage infiltration rate of the plaque were significantly decreased by the higher dose of curcumin phytosomes compared to the lower one, the measurement being performed on hematoxylin-eosine stained sections of the aortic arch [152].

The group of Poruba M. et al. [153] administered silymarin in different formulations (a standardized extract of silymarin, micronized silymarin and silymarin in the form of phytosome) to non-obese hereditary hyper-triglyceridemic rats as dietary supplements (1%) for 4 weeks. The results showed that all tested forms of silymarin significantly decreased the plasma levels of TG, TC and increased the HDL-C levels [153]. Silymarin formulated in phytosomes significantly increased the protein expression of cytochrome P450 4A (CYP4A), known as contributing to omega and omega-1 hydroxylation of fatty acids required for the synthesis of TG [153]. The silymarin in the form of phytosome had the best bioavailability [154]. All the three silymarin diets significantly increased the protein expression of cholesterol 7alpha-hydroxylase (CYP7A1), a hepatic enzyme responsible for metabolizing cholesterol into 7α-hydroxycholesterol, a rate-limiting step for the bile acids synthesis. Silymarin in the form of phytosome significantly increased the protein expression of ATP-binding cassette (ABC) transporters (ABCG5 and ABCG8), regulating the cholesterol efflux from the hepatocytes into the bile [153]. The positive effect of silymarin on plasma TC could be partly due to the higher excretion of cholesterol mediated by CYP7A1 and ABC transporters. The decreased plasma level of TG may be due to the increased liver TG metabolism through CYP4A. In this case, the formulation in phytosomes added therapeutic value to silymarin.

#### 4.2.2. Clinical Studies 

Bergamot (*Citrus bergamia*) offers a unique profile of flavonoids and flavonoid glycosides, such as neoeriocitrin, neohesperidin, naringin, rutin, neodesmin, rhoifolin and poncirin [155]. Rondanelli M. et al. [156] aimed to evaluate the effect of bergamot phytoconstituents formulated in phytosomes on the lipid parameters in 64 overweight and obese class I subjects (BMI 25–35 kg/m^2^) with diagnosed mild hypercholesterolemia (5.4–7.0 mmoL/L). Bergamot Phytosome (Vazguard, Indena SpA, Milan, Italy) is an innovative lecithin formulation of the bergamot enriched polyphenols fraction (BPF), the phospholipids (sunflower lecithin) being formulated with 40% in weight standardized BPF extract as described previously by Mollace V. et al. [93], in order to enhance the oral bioavailability of bergamot’s main flavonoids. Tablets contain 500 mg of bergamot phytosomes (standardized to contain 11–19% of total flavanones). The bergamot phytosomes were administered for 12 weeks to overweight and obese subjects, and after the first 30 days of treatment the values of TC and LDL-C significantly decreased, while the level of HDL-C increased compared to the placebo group [156]. The complex composition of bergamot extract endowed the phytosomes with multiple mechanisms of action. Thus, the signaling pathways involved the direct stimulation of AMP-activated protein kinase (AMPK), and the inhibition of HMG-CoA reductase by some flavanones that have a hydroxyl mevalonate moiety [156].

### 4.3. Hepatic Disorders

The liver works like a central factory for the body metabolism, being the organ responsible for nutrients processing, synthesis and catabolism of numerous key proteins and lipids, as well as for drugs detoxification. Therefore, the health of the liver is very important for the health of the whole body. NAFLD or the metabolic-associated fatty liver disease (MAFLD) affects over 30% of the world population [157]. It arises from lipid accumulation in hepatocytes, a process that alters the liver’s normal function. It is associated with obesity, dyslipidemia and insulin resistance, which are risk factors for type 2 DM (T2DM) and CVD [157,158]. Of interest, 75% of T2DM patients also display NAFLD [157]. To date, there is no specific treatment for NAFLD, but previous studies have shown that diets rich in antioxidants and anti-inflammatory phytochemicals, with little or no side effects, can be effective in treating NAFLD [159].

#### 4.3.1. Preclinical Studies

The Ginkgoselect Phytosome^®^ (Indena SpA, Milan, Italy), prepared from soy phospholipids and *Ginkgo biloba* extract, has been tested for hepatoprotective properties in Wistar rats with rifampicin-induced hepatotoxicity (500 mg/kg, for 30 days). Treatment with Ginkgoselect Phytosome^®^ at 25 mg/kg and 50 mg/kg significantly lowered the plasma levels of alanin aminotransferase (ALT) and aspartate aminotransferase (AST) and increased the serum total proteins and albumin, the latter being indicators of the hepatoprotective effects. No differences between the groups treated with phytosome and those treated with 100 mg/kg silymarin were observed [96]. Similar, the group of Naik S.R. et al. [160] conducted a study on Wistar albino rats with carbon tetrachloride (CCl_4_)-induced liver damage and intraperitoneally treated with *Ginkgo biloba* phytosomes (25 mg/kg and 50 mg/kg) before and during the CCl_4_ treatment. The results show that the serum activity of ALT, AST and alkaline phosphatases (ALPs) were significantly decreased in a dose-dependent manner, while the serum total protein and albumin levels were increased in the phytosome-treated group compared to the untreated animals [160]. The hepatoprotective effect of *G. biloba* phytosomes has been confirmed by a histopathologic assay showing the regeneration of the liver cells after CCl_4_-induced injury and centrilobular necrosis of the liver tissue. The beneficial effects of phytosomes have been comparable to those of silymarin, a known hepatoprotective drug. In addition, the *G. biloba* phytosomes treatment increased the hepatic level of GSH and the activity of the antioxidant enzymes SOD, CAT, GPx and GR, while the TBARS levels were decreased in a dose-dependent manner, compared to untreated animals [160].

*Silybum marianum*, known as milk thistle, is a recognized hepato-protectant used in the treatment of hepatitis C, hepatocarcinoma, NAFLD and gall bladder disorders [161,162,163]. Four isomers have been described within silymarin, such as silybin, isosilybin, silicristin and silidianin, the main active component being silybin. Like many other naturally occurring compounds, silymarin has limitations such as partial water-solubility, poor bioavailability and poor intestinal absorption. In order to increase the bioavailability and to enhance the beneficial effects of milk thistle extract, El-Gazayerly O.N. et al. [164] prepared phytosomes containing different molar ratios of soy lecithin/egg yolk and silybin, but the most suitable formula was that which contained 0.25:1 egg yolk and silybin. Milk thistle extract (equivalent to 200 mg silybin/kg, orally) and silybin phytosomes (equivalent to 200 mg/kg silybin, orally) were administered daily to male albino rats with CCl_4_-induced liver injury for 10 days. Both the phytosomes and the milk thistle extract induced elevated SOD levels in the treated groups, but the phytosomes were more efficient than milk thistle extract in this respect. The phytosomes also proved to be more effective compared to the milk thistle extract in the case of ALT, whose level was more decreased in the phytosomes group [164]. In another study, Shriram R.G. et al. [47] investigated the efficacy of phospholipid-based sylimarin phytosomes in enhancing the absorption and the oral bioavailability of silymarin, as well as their hepatoprotective properties, in male Wistar rats. The animals were orally treated by an optimized silymarin phytosomal suspension (100 mg silymarin/kg/day) for 7 days, followed by a single i.p. dose of a mixture of CCl_4_ and olive oil on the seventh day. The silymarin phytosomal formulation significantly improved silymarin oral bioavailability compared to pure silymarin, as indicated by a 6-fold increase in the systemic bioavailability. The results show that the treatment with silymarin phytosomes was most efficient in restoring liver SOD, CAT, GPx, glutathione S-transferase (GST), GR and GSH levels in rats, the increase in these enzymes’ levels being more evident than in the group treated with plain silymarin. Pre-treatment with silymarin–phospholipid complexes significantly abrogated the CCl_4_-induced increase in lipid peroxides measured as MDA levels, while pre-treatment with pure silymarin failed to protect the rats from CCl_4_-triggered lipid peroxidation. Pre-treatment with pure silymarin resulted in a moderate hepatoprotective effect as evidenced by a decrease in fatty tissue degeneration and parenchymal cells damage. The silibinin accounts for 50–70% of the milk thistle extract, and it is the major bioactive flavonolignan in silymarin. Tang S. et al. [165] assessed the hepatoprotective mechanism of silibinin–phospholipids complex in the pathogenesis of acute liver injury and investigated how silibinin phytosomes modulate necroptosis-S100A9-necroinflammation signaling molecules. To achieve this, BALB/c mice treated with D-galactosamine (D-GalN)/LPS (in order to trigger liver injury) received by gavage silibinin (25 mg/kg) or silibinin–phospholipid complex (25 mg/kg) 24 h before and 2 h after acute liver insult. Serum activity of AST and ALT, as well as the hepatic pathology score, were significantly decreased in the unformulated and formulated silibinin-treated groups compared to the injured group, but there was no significant difference between free silibinin and phospholipid complex formulation treatment. Of interest, the silibinin–phospholipid complex treatment decreased the levels of necroptosis-signaling molecules, S100 calcium-binding protein A9 (S100A9) and necroinflammation signaling molecules, making it more effective compared to unformulated silibinin. In addition, the authors incubated macrophages isolated from the tibia and femur of mice with formulated silibinin, in order to evaluate whether silibinin–phospholipid complex treatments promote the polarization of macrophages toward an anti- or pro-inflammatory phenotype. Interestingly, they detected a reduction in M1 markers (such as inducible nitric oxide synthase-iNOS, CD86 and TNFα) and an increase in M2 markers (such as arginase 1-Arg1, CD206 and transforming growth factor-beta-TGF-β), the level of TGF-β being significantly higher in cells exposed to silibinin phytosomes compared to unformulated silibinin. This can be a mechanism responsible for the hepatoprotective effects of phytosomes formulation of silibinin, and all these results prove the advantages of this formulation for the therapeutic potential of silymarin.

Naringenin is a natural bioactive compound belonging to the flavanone class, possessing antioxidant, anti-inflammatory and lipid-regulatory effects [19]. Due to its rapid elimination, naringenin needs frequent administration to maintain an effective plasma level. The group of Mukherjee P.K. [166] prepared naringenin–phospholipid complexes and evaluated their antioxidant and hepatoprotective properties compared to free naringenin at a dose of 100 mg/kg in the model of CCl_4_-injuried rats. Their results indicated that formulation of naringenin in phytosomes increases its maximum concentration by over 60% and prolongs its life in circulation (more than double, from 10 h at maximum concentration to 25 h), compared to free naringenin. Thus, the phospholipid formulation of naringenin increases its half-life and decreases the clearance rate, making it more effective in protecting the liver against harmful CCl_4._ In this manner, the pre-treatment of rats with naringenin phytosomes significantly reduced the plasma concentration of ALT, AST and bilirubin increased by CCl_4_ administration, and increased the hepatic levels of the antioxidant enzymes GPx, SOD and CAT, compared to free naringenin which has no significant beneficial effects. 

Catechin is a natural bioactive compound belonging to the flavonoids class that has been shown to be a ROS scavenger, a promoter of anti-oxidant enzymes, hepatoprotective and inhibitor of pro-oxidant enzymes [167,168]. The group of Athmouni, K. et al. [169] developed catechin–phospholipid complexes and evaluated their protective effect on cadmium-caused liver injuries in rats. They found that catechin (15 mg/kg body weight) absorption in rats increased by 40% following formulation in phytosomes, compared to the unformulated compound, and this could be related to its increased solubility. In addition, treatment of rats having cadmium-induced liver injuries with catechin phytosomes induced a significant decrease in plasma AST, ALT and bilirubin, and ameliorated the activities of the antioxidant enzymes SOD, CAT and GPx, compared to unformulated catechin.

Apigenin is a hydrophobic, polyphenolic flavonoid known for its antioxidant, anti-inflammatory, antidiabetic and anti-microbial potential [19]. The group of Telange D.R. et al. [43] developed apigenin–phospholipid phytosomes to improve the aqueous solubility, in vivo bioavailability and antioxidant activity of apigenin. The optimized formulation demonstrated a 36-fold higher aqueous solubility of apigenin, and a significant increase (over 40%) in the bioavailability up to 8–10 h, compared to that of free apigenin (at a dose of 100 mg apigenin/kg). The hepatoprotective role of apigenin phytosomes was assessed on the rat model of CCl_4_-induced liver injury, and the results show that phytosomes containing 25 mg apigenin/kg significantly increased the levels of GSH, SOD CAT, and decreased the levels of lipid peroxides (TBARS), compared to free apigenin.

Ursolic acid is a pentacyclic triterpenoid, which is found either as a free acid or as an aglycone of saponins. It is reported to have several therapeutic activities including antihyperlipidemic effects by inhibiting the activity of pancreatic lipase [170] and it is also known for its hepatoprotective potential in both acute and chronic liver diseases [171]. The major limitations of ursolic acid are its poor absorption, rapid elimination and hence low bioavailability. Therefore, the group of Biswas S. [72] prepared and characterized phospholipid complexes of ursolic acid in order to overcome these limitations. They investigated ursolic acid’s hepatoprotective activity and bioavailability in CCl_4_-injuried Wistar rats. The animals were treated with free ursolic acid extract or formulated in phytosomes at doses of 10 and 20 mg/kg p.o. equivalent to pure ursolic acid, for 7 days. After this time, a single i.p. dose of a mixture of CCl_4_ and olive oil was administered to rats, to induce liver injury. The results show that the phytosome formulation increased the serum bioavailability of ursolic acid by over 8-fold, and enhanced its elimination time by 12-fold as compared to the pure compound at the same dose. Both formulas had hepatoprotective actions in CCl_4_-treated rats demonstrated by the decrease in serum AST, ALT and ALP activity, the effects being slightly increased in the phytosome group compared to those with free ursolic acid. Total bilirubin serum levels were significantly decreased only in the group that received ursolic acid formulated in phytosomes, compared to the CCl_4_ -treated group. The activity of the liver antioxidant enzymes SOD, CAT, GPx, GST and GR, the GSH levels and the total protein concentration in the liver were best stimulated by the administration of 20 mg/kg ursolic acid formulated in phytosomes. In addition, the architecture of the liver tissue was significantly improved by the administration of phytosomes with ursolic acid, compared to pure ursolic acid or to the CCl_4_ -treated group, as it was evidenced by the hematoxylin-eosin-stained sections. 

All these preclinical studies bring solid arguments for the use of phytosomal formulation of bioactive compounds to treat hepatic disorders.

#### 4.3.2. Clinical Studies

In the study conducted by Safari Z. et al. [172], patients with NAFLD received curcumin phytosomes (250 mg containing 20% curcuminoids and 20% phosphatidylserine, Indena SpA) or placebo pills daily for 12 weeks, followed by the evaluation of the lipid profile, fasting blood sugar, anthropometric indices, liver enzymes, fibrosis and steatosis. The results showed that the administration of phytosomal curcumin significantly improved the liver status by reducing fibrosis and steatosis in NAFLD patients compared to the placebo group, but there was no difference in the lipid profile (TC, TG, LDL-C, HDL-C) and fasting blood glucose between these groups [172]. Panahi Y. et al. [173] aimed to evaluate the efficacy and safety of supplementation with phytosomal curcumin (Meriva^®^, Indena SpA, Milan, Italy), which contained a complex of curcumin and soy phosphatidylcholine in a 1:2 weight ratio, in subjects with NAFLD. The administration of phytosomal curcumin (1000 mg/day in 2 divided doses) for 8 weeks to NAFLD patients induced a significant decrease in the body mass index (BMI) values, the lowering of hepatic AST and ALT activities as well as the reduction in the portal vein diameter and liver size as compared to the placebo group. An increase in the hepatic vein flow velocity was also observed between the two analyzed groups. Furthermore, ultrasonographic findings (decrease in the echogenicity of the liver parenchyma) were improved in 75% of subjects in the curcumin group, while the rate of improvement in the placebo group was 4.7%. Formulation into nanophytosomes containing piperine has been shown to improve the low bioavailability of curcumin by decreasing the conjugation of curcumin with glucuronic acid in the liver and consequently lowering its elimination in urine [138]. Using these types of nanophytosomes, Cicero A.F.G. et al. [174] conducted a study in which they administered curcumin-containing phytosomes (Curserin^®^, Indena SpA, Milan, Italy: 200 mg curcumin, 120 mg phosphatidylserine, 480 mg phosphatidylcholine and 8 mg piperine from *Piper nigrum* L. dry extract) for 8 weeks to overweight subjects with suboptimal fasting plasma glucose. The results showed that the treated group exhibited improved lipid profile (as demonstrated by the decrease of TG and increase of HDL-C levels) as compared to the placebo group. In addition, it was noticed the improvement of the hepatic function, reflected by the reduction in the activities of AST, ALT and gamma–glutamyl transferase, as well as in the fatty liver score [174]. 

Larger clinical studies are needed to consolidate the therapeutic potential of curcumin formulated in phytosomes for hepatic disorders.

### 4.4. Diabetes Mellitus

Diabetes is one of the 21st century’s major health concerns, being one of the most encountered disorders nowadays, with a more than 50% increase since 2017 [175]. The alteration in the microvasculature or macrovasculature leading to cardiovascular problems, diabetic kidney disease, diabetic neuropathies or retinopathies determines increased morbidity and mortality in diabetic patients [175]. The hallmark of DM is the increased level of plasma glucose which appears as a consequence of an altered insulin metabolic pathway, either due to its decreased secretion (caused by the dysfunction or even loss of function of pancreatic beta cells, characteristic to type 1 DM), defective action (insulin resistance, characteristic to T2DM) or both [8]. Another characteristic of DM is the presence of dyslipidemia characterized by elevated fasting and postprandial TG, increased LDL-C levels and low levels of protective HDL-C, which also plays an important role in the development of diabetic vasculopathies [8]. Synthetic oral anti-hyperglycemic drugs such as insulin, sulfonylureas, thiazolidinediones and biguanides were successfully developed for diabetes treatment. Unfortunately, many of these drugs have significant side effects such as fatal hepatotoxicity, increased risk of myocardial infarction and increased cardiovascular mortality (reviewed in [14]). To overcome these problems, alternative or complementary therapeutic strategies with little side effects, such as phytotherapy, were developed [139]. Different complex mixtures or classes of bioactive compounds including alkaloids, flavonoids, polyphenols and triterpenes were used for the amelioration of diabetic symptoms.

#### 4.4.1. Preclinical Studies

Rani A. et al. [51] tested phytosomes containing curry tree (*Murraya koenigii*) extract to assess their antidiabetic effects in vivo, using STZ-induced diabetic Wistar rats. The animals were treated for 21 days with glucose and with different doses of extract (200–400 mg/kg) or phytosomes (100–200 mg/kg). The animals treated with phytosomes exhibited a 40% reduction in serum glucose concentration at a lower dose, suggesting enhancement in its therapeutic efficacy compared to the group treated with the extract. The levels of urea and creatinine, important markers for renal disorder in DM, were restored to almost normal levels in the treated group, indicating an antidiabetic activity of curry tree phytosomes [51].

*Crataegus aronia* is used to treat DM [50]. Nanophytosomes encapsulating *C. aronia* leaf extract were developed very recently by the group of Altiti A. et al. [50] by mixing 1:1 phospholipid (lecithin) with *C. aronia* leaf ethanolic extract. The nanophytosomes (150 mg/kg or 250 mg/kg) had hypoglycemic effects at 14 and 21 days of treatment of STZ-induced diabetic Wistar albino rats. The effect was dose-dependent, slightly more efficient than that of the unformulated extract at the same doses, but not equipotent to metformin [50].

An interesting therapeutic approach designed to increase the beneficial effects of natural compounds was that in which a complex of more than one natural extract is encapsulated in the same phytosome. This approach was chosen by Rathee S. and Kamboj A. [176] who used extracts from fruits of *Citrullus colocynthis* (L.), *Momordica balsamina* and *Momordica dioica* to obtain antidiabetic phytosomes. The optimized formula for these phytosomes was obtained by using the three-level Box–Behnken design which considers parameters such as phospholipid: extract ratio, process temperature and reaction time. The resulting phytosomes were stable and had a very good entrapment efficiency, while the polyherbal extracts and phospholipids in the complex were joined by non-covalent-bonds and did not form a new compound. To test the anti-diabetic effects, the polyherbal phytosomal formulation was administered to STZ-induced diabetic Wistar rats in doses of 100 mg/kg and 250 mg/kg for 15 days. The results show that, although both concentrations are effective in decreasing the serum glucose level, the lower concentration is more efficient, its effect being comparable to that of 50 mg/kg metformin. 

Chrysin is a flavonoid encountered in different plants, flowers and bee propolis. In the context of DM, in vivo studies demonstrated that chrysin has hypolipidemic, anti-inflammatory and anti-oxidant effects [177,178,179]. However, the decreased bioavailability of chrysin due to its poor solubility and short circulation half-life leads to the necessity to formulate it in an efficient delivery system. Recently, the group of Kim S.M. et al. [179] used chrysin-loaded phytosomes prepared with egg phospholipid at a 1:3 molar ratio (chrysin: phospholipid) to study the antidiabetic effects of this natural compound. After 9 weeks of nanophytosomes administration (100 mg chrysin equivalent/kg) to db/db mice, the study demonstrated a decrease in serum glucose, insulin levels and homeostatic model assessment for insulin resistance (HOMA-IR), similar to 200 mg/kg metformin treatment. The hypoglycemic effect of chrysin nanophytosomes was due to their capacity to inhibit the gluconeogenesis by down-regulating phosphoenolpyruvate carboxykinase (PEPCK) in the liver, while stimulating the glucose uptake in db/db mice by increasing glucose transporter type 4 (GLUT4) gene expression and translocation in the skeletal muscle. Compared to free chrysin, the chrysin nanophytosomes demonstrated an additional effect in modulating the glucose metabolism-related genes both in the liver (PEPCK, hexokinase 2) and skeletal muscle (GLUT4, hexokinase 2 and PPARγ) due to the increased bioavailability of the chrysin in the phytosome formulation [179].

Rutin is a citrus flavonoid glycoside found in different plants, including *Ruta graveolens* L. (Rutaceae), *Eucalyptus* spp. (Myrtaceae) or *Sophora japonica* L. (Fabaceae) [180]. Besides being a powerful antioxidant and anti-inflammatory agent, rutin is also known for its antidiabetic action. The mechanisms of rutin anti-diabetic effects include the reduction in carbohydrates absorption from the small intestine, the up-regulation of glucose uptake paralleled by the suppression of gluconeogenesis as well as the increase in insulin secretion from pancreatic beta cells [181]. To overcome the limitation of its low bioavailability, rutin was encapsulated by Amjadi S. et al. [182] into nanophytosomes in order to evaluate the therapeutic potency of this nanocarrier in STZ-induced diabetic Wistar rats. The administration of rutin-loaded nanophytosomes (25 mg rutin/kg per day) for 4 weeks to diabetic rats decreased the level of glucose and glycated hemoglobin in the blood and restored the diabetes-induced damages in the pancreas, liver and kidney (evidenced by the histopathological assay) in a more efficient manner than free rutin. This demonstrates that encapsulation into phytosomes is an efficient technique to enhance the rutin therapeutic potential in diabetic conditions. In addition, rutin nanophytosomes were more effective than free rutin to control the hyperlipidemia, measured as decreased TC, TG and increased HDL-C, to reduce the AST and ALT activities, and to attenuate the oxidative stress expressed as decreased MDA levels and increased total antioxidant potential due to the increased activity of SOD and GPx in the animal model. 

Curcumin is a polyphenol with well-known anti-diabetic, lipid-lowering and hepatoprotective actions [183,184]. Studies have shown that curcumin possesses insulin-sensitizing actions through different pathways. It was demonstrated that the molecular mechanisms of action of curcumin in DM include the activation of insulin receptors, the increase in lipoprotein lipase activity, the stimulation of insulin-independent glucose uptake by pancreatic beta cells and anti-inflammatory effects in adipose tissue, in parallel with the enhancement of adipokines [139]. Recently, the group of Alhabashneh W. et al. [185] tested curcumin phytosomes effects on glycemic and lipid profile of STZ-induced Wistar diabetic rats. The phytosomes obtained by mixing 1:1 phosphatidylcholine (soybean lecithin) and curcumin were orally administered to diabetic rats at 150 and 250 mg/kg dose, for 3 weeks. The results show that curcumin phytosomes administration determines the decrease in blood glucose, TC, LDL-C and TG in parallel with the significant rise in HDL-C. The hypoglycemic and hypolipidemic effects were dose-dependent and the effects of curcumin phytosomes were higher than for those treated with free curcumin at the same dose [185]. 

Berberine is a multi-target natural product with beneficial effects in different pathologies. Studies in humans show that berberine has a beneficial effect on the expression of genes which ameliorate glucose levels (1 mg/day), regulate the cholesterol absorption (300 mg/day) or modulate the microbiota (500 mg/day) [186]. In the context of DM, berberine has positive effects due to is antioxidant properties and also, very importantly, because it targets AMPK, a protein involved in the regulation of glucose metabolism, fatty acid oxidation and insulin resistance [187]. To overcome the low bioavailability of free berberine, the group of Yu F. et al. [52] developed phytosomes loaded with berberine using soybean phosphatidylcholine and commercially available berberine in a mass ratio of 5:1. Pharmacokinetic studies demonstrated that the oral bioavailability of the berberine-loaded nanophytosomes in Wistar rats was improved 3-fold as compared to the orally administrated free berberine. The obtained phytosomes have also anti-diabetic effects, the study demonstrating that administration of berberine in nanophytosomes (100 mg/kg) for 4 weeks to diabetic db/db mice decreased the plasma fasting blood glucose and reverted the TG levels in the liver to a better extent than free berberine. The insulin levels were not modulated either by free berberine or by the phytosome formulation.

*Gymnema inodorum* was successfully used in alternative medicine for the amelioration of DM. Its antidiabetic properties are due to the presence of active phenolic acids, flavonoids, triterpenoid compounds and pregnane glycosides, this complex composition conferring the *G. inodorum* extract antiglycemic and antioxidant effects and insulin-mimetic properties [121,188,189,190]. The *G. inodorum* leaf extract formulated in phytosomes has been tested for its anti-insulin resistance activity by measuring the glucose uptake and lipolysis in LPS-treated 3T3-L1 adipocytes. The results evidence that LPS adipocytes exposed to phytosomes with *G. inodorum* extract take up more glucose compared to LPS adipocytes alone, but higher levels of phospholipid in phytosomes (mass ratio GI extract: PC = 1:2 versus 1:1) slightly interfered with the anti-insulin-resistant effects of the *G. inodorum* extract by decreasing the glucose uptake and increasing the lipid degradation, measured as glycerol release in the culture medium [26]. These data highlight that an increased percentage of phospholipids in phytosomes can invalidate their beneficial effects through the oxidation products they can generate.

Tripterine is a pentacyclic triterpenoid quinone with anti-diabetic and anti-inflammatory potential. The group of Zhu S. et al. [191] used selenized tripterine phytosomes (Se@Tri-PTs) to identify the mechanisms by which these nanostructures alleviate podocyte injury, a major contributor to diabetic nephropathy. The study showed that 5 µg/mL of selenized phytosomes improved the viability of podocytes in vitro. In addition, the phytosomes reduced NLRP3 expression, re-established the expression of proteins that regulate autophagic processes (Beclin-1, microtubule-associated protein 1A/1B-light chain 3-LC3 II/LC3I, ubiquitin-binding protein p62-p62 and SIRT-1) and decreased the apoptosis of podocytes exposed for 48 h to high glucose [191].

#### 4.4.2. Clinical Studies

Quercetin is a flavonol found in various fruits, vegetables or leaves, with beneficial effects in human health. To overcome quercetin’s highly variable bioavailability (0–50%) and rapid elimination (1–2 h half-life time), different innovative formulations to deliver this active compound were tested. Quercetin Phytosome^®^ (Quercefit™, Indena SpA, Milan, Italy) is one of the quercetin formulations that demonstrated improved solubility in the solution that simulated gastrointestinal fluids and also increased absorption (20-fold) [192]. Using this formulation, Riva A. et al. [192] evaluated the interaction between Quercefit and anti-diabetic therapy, antiplatelet agents and anticoagulants, in healthy subjects. The topic is important, due to the fact that it is well known that the interaction between synthetic drugs with different natural active compounds exists, and it can have serious clinical outcomes [193,194]. The study involved 12 diabetic patients treated with antidiabetic metformin, 30 subjects with antiplatelet therapy (acetylsalicylic acid, ticlopidine or clopidogrel) and 20 subjects with anticoagulants (warfarin or dabigatran), all of them receiving supplementation with Quercetin Phytosome for at least 10 days. The Quercefit supplementation consisted in the oral administration of two tablets of phytosomes/day, corresponding to 200 mg/day of quercetin. The results showed that Quercefit administration did not induce significant differences in the fasting glycaemia or glycated hemoglobin (for the anti-diabetic medication), measured bleeding time (for subjects with antiplatelet therapy) or International Normalized Ratio (INR) level (for the subjects with anticoagulant therapy) [178]. This study proves that the administration of quercetin phytosome does not interfere with different classes of allopathic drugs, such as metformin, aspirin or warfarin.

Meriva curcumin phytosomes (Indena) were also used to evaluate the possible beneficial actions of curcumin to improve the kidney functions of subjects with temporary kidney dysfunction (TKD). This study included a total of 87 subjects with TKD divided into two groups: one following the standard management (hydration, a reduced intake of NaCl and other electrolytes, decreased intake of carbohydrates, abolition of drugs or compounds that potentially harm the kidneys and a moderate exercise program) and a group that followed the same standard management and received also a supplementation with Meriva^®^ (1.5 g/day, delivering 300 mg highly bioavailable curcumin) divided into three administrations/day, for 4 weeks. The results of the study show that Meriva^®^ reduced the albumin almost to normal levels in the urine of patients with microalbuminuria and macroalbuminuria, and decreased the oxidative stress measured as plasma free radicals. The tolerability to Meriva^®^ and the compliance of the patients were good. These parameters were improved compared to those from the standard management group, suggesting that Meriva can safely ameliorate the TKD symptoms [195].

The group of Cicero A.F.G. et al. [174] recently studied the effect of curcumin phytosome containing piperine (Curserin^®^, Indena, SpA, Milan, Italy) on a placebo-controlled clinical trial including 40 overweight subjects who received Curserin^®^ compared to 40 overweight subjects that received a placebo. The study showed that two capsules/day of Curserin^®^ administration for 8 weeks have beneficial effects on the: (i)anthropometric parameters (decreased BMI and waist circumference) and (ii) metabolic characteristics (decreased HOMA-IR, fasting plasma glucose and fasting plasma insulin) as compared to the placebo group. Very interesting, the authors observed for the first time a statistically significant reduction in serum cortisol level in human subjects treated with Curserin^®^. In another study, Mirhafez S.R. et al. [196] investigated the therapeutic properties of lower doses of phospholipid formulation of curcumin on the lipid profile, hepatic enzymes and hepatic fat mass in patients with NAFLD in a randomized controlled clinical trial. Patients in the treated group received curcumin phytosome in a dose of 250 mg/day (Meriva, Indena SpA, Milan, Italy) for 2 months. The results show that AST levels in the serum of the phytosome-treated group were significantly lower, as well as the NAFLD grade, compared to the placebo group. 

Berberine phytosome formulation (Indena SpA, Milan, Italy), containing berberine extract (28–34%), in combination with sunflower lecithin, pea protein and grape seed extract, was used in a recent pilot study to evaluate their possible beneficial effects in polycystic ovary syndrome (PCOS), a medical condition associated with insulin resistance. A group of 12 normal and overweight women received two daily oral doses of 550 mg berberine tablets for 60 days. The treatment determined the normalization of (i) the glycemic and insulin profiles, measured as decreased HOMA-IR index, insulin and glycemia; (ii) lipid profile, evidenced as a lowering of VLDL-C and TG levels and (iii) inflammatory status measured as a decrease in TNFα and CRP levels in the plasma. In addition, the authors observed the redistribution of adipose tissue with the reduction in the visceral fat and fat mass, without any changes in the diet [187]. Importantly, the tested product was well-tolerated, with no modifications in safety blood parameters as AST, ALT, gamma–glutamyl-transferase or bowel discomfort, the lack of side effects being very important, permitting the long-term use of berberine.

### 4.5. Metabolic Syndrome 

The metabolic syndrome represents a condition including a cluster of risk factors that contribute to the inception and progression of cardiovascular diseases. The risk factors are dyslipidemia (high TC, LDL-C and TG levels, low HDL-C levels), abdominal obesity, impaired fasting glucose/insulin resistance and high blood pressure. Among the induced diseases, obesity is a complex disorder involving the excessive accumulation of fat in the body. It is the most common disorder worldwide, which leads to many complications such as DM, stroke and CVD. The above complications represent the leading causes of mortality worldwide [197]. The treatment of obesity is mostly based on equalizing calorie ingestion and energy consumption. Several natural compounds are used for the treatment of obesity.

#### 4.5.1. Preclinical Studies

*Callistemon citrinus* has been described as antimicrobial, anti-inflammatory, anti-obesogenic, antioxidant and hepatoprotective [198,199]. The main bioactive components are terpenoids, phenolic acids and flavonoids (including eucalyptine, blumenol, gallic acid and protocatechuic acid) which have low oral bioavailability and absorption. The group of Ortega-Pérez L.G. et al. [200] prepared *Callistemon citrinus* leaf extract in phyto-phospholipid complexes aiming to use them (doses range 50–200 mg extract/kg) for weight gain prevention in Wistar rats fed with a hypercaloric diet. They obtained phytosomes with the *C. citrinus* extract of a small size, high entrapment efficiency, improved oral bioavailability and enhanced stability at 20 °C (over three months) [201,202]. The phytosomes and unformulated *C. citrinus* extract exhibited similar inhibitory activity against the free radicals assessed by 2,2-diphenyl-1-picrylhydrazyl (DPPH) and 2,2′-azino-bis(3-ethylbenzothiazoline-6-sulfonic acid) (ABTS) methods, and the ability to reduce ferric to ferrous ions. In addition, both products reduced excessive weight in the obese rats. The administration of *C. citrinus* phytosomes, even in low doses, further reduced the adiposity index and TG levels in obese rats, compared to the unformulated extract [200]. 

Soybean (*Glycine max*) is known to influence body weight due to its content of saponin, proteins and phosphatidylcholine [203,204]. The topical route to administer bioactive compounds has received considerable attention because it is a non-invasive path and is characterized by high bioavailability due to the direct drug delivery to the site of action, avoiding losses along the gastrointestinal tract [205]. Thus, the group of El-Menshawe S.F. et al. [69] investigated the anti-obesity effect of soybean extract formulated in phytosomes and included them into a thermogel to be applied on the abdomen of experimental male albino rats fed with high-fat diet in order to gain weight. Their results showed that, after a one-month treatment, the soybean phytosomal thermogel induced a decrease in body weight (10%), adipose tissue weight (27%) and food consumption (35%) compared to rats receiving thermogel with unformulated soybean extract. In addition, the animals with soybean phytosomal thermogel presented a significant decrease in plasma levels of TC, TG, LDL-C and VLDL-C, evidencing a slight systemic action of the formulated extract included in thermogel. Moreover, the animals treated with soybean phytosomal thermogel have shown a decreased size in the adipose cells in the epididymal adipose tissue, compared to the crude soy thermogel group [69]. Therefore, topical phytosomal application of plant extracts could be considered a promising formulation for future treatments.

#### 4.5.2. Clinical Studies

The phyto-phospholipid formulation of curcumin was tested in human subjects diagnosed with metabolic syndrome to evaluate its ability to increase the vitamins levels, such as vitamin E. Thus, the group of Mohammadi A. et al. [206] conducted a study that involved 120 subjects with metabolic syndrome, part of them receiving unformulated curcumin, the others receiving lecithinized curcumin for 6 weeks in a dose of 200 mg active substance/day. The results evidenced that neither unformulated curcumin, nor the lecithinized form, have an effect on serum levels of vitamin E, suggesting that the antioxidant properties of curcumin are not driven on vitamins, but on the modulatory effects exerted on the antioxidant enzymes which further induce anti-inflammatory effects discussed in Section 4.1.2. 

The group of Rondanelli M. et al. [156] evaluated the effect of bergamot phytoconstituents formulated in phytosomes (Bergamot Phytosome, Vazguard, Indena SpA) on visceral adipose tissue, as an indicator for the metabolic syndrome, DM and CVD [207,208] in overweight and obese class I subjects. After one month of treatment with bergamot phytosomes, the subjects showed a significant reduction in visceral adipose tissue compared to the placebo group [156]. These results are of interest because visceral adipose tissue is considered an endocrine organ able to influence the function of other organs such as liver, heart or blood vessels. Beside the lipid-lowering effects of bergamot formulated in phytosomes described in Section 4.2.2, its action as bodyweight regulator increases its therapeutic potential. 

The capacity to induce weight loss of Greenselect Phytosome, a green tea extract devoid of caffeine and formulated in lecithin to improve the absorption of catechins, was evaluated in a single blind, controlled study on 50 asymptomatic subjects with borderline metabolic syndrome factors and with increased plasma oxidative stress. A group of 50 similar volunteers who took the blank formulation was considered as control group. Greenselect Phytosome in the form of coated tablets (150 mg/tablet) was administered for 24 weeks, and the results show that the phytosome formulation promoted weight loss (8%), reduced waist circumference (6%) and decreased the value of plasma free radicals by 33%, as compared to the control group [83]. This study confirms the efficacy of green tea extract formulated in phytosomes in inducing weight loss.

### 4.6. Cardiovascular Disorders

CVD represent the main cause of morbidity and mortality worldwide, and hypercholesterolemia, hypertension, obesity, left ventricular hypertrophy or DM represent common CVD risk factors [209]. CVD includes disorders such as heart failure, myocardial infarction (MI), pulmonary embolism and stroke [210]. The molecular mechanisms of CVD development have been extensively investigated in recent decades, the oxidative and inflammatory stress being identified as important players in the advancement of these disorders. Although important progresses were made, some of the current pharmacological therapies used for CVD treatment (including statins, antihypertensive, antithrombotic and anti-coagulation agents) present important side effects, the compliance of patients to the therapy being sometimes reduced. In the last decade, natural products based on concentrated mixtures of bioactive compounds with low side effects compared to pharmacological therapies were developed and experimentally evaluated in order to evaluate their potential to ameliorate CVD effects.

#### 4.6.1. Preclinical Studies

*Gymnema sylvestre* has been reported as possessing antioxidant, hypolipidemic and antidiabetic properties, which have been attributed to its bioactive compounds, such as triterpene glycoside named gymnemic acid [211,212]. The cardioprotective potential of phospholipid–gymnemic acid complexes was evaluated in a rat model of cardiomyopathy induced by doxorubicin (30 mg/kg/i.p./single dose). Pre-treatment with gymnemic acid phytosomes (50 and 100 mg/Kg/p.o./day) for 30 days significantly reduced the cardiac toxicity induced by doxorubicin, including improvement in hemodynamic parameters and the ratio of heart weight to body weight. Moreover, the phytosome treatment decreased the concentration of Ca^2+^ and lactate dehydrogenase (LDH) in serum, as well as the levels of caspase-3 and TBARS in myocardium, compared to the untreated group. In addition, the gymnemic acid formulated in phytosomes increased the levels of Na^+^/K^+^ ATPase and antioxidant enzymes SOD, CAT and GPx, as compared to the pathogenic control group. The anti-apoptotic effect of phytosomes with gymnemic acid was confirmed by prevention of inter-nucleosomal DNA laddering on agarose gel electrophoresis [213]. 

The extracts from the leaves of *Ginkgo biloba* have been found to possess cardioprotective, antioxidant, hepatoprotective and antidiabetic properties [214,215]. The antioxidant and cardioprotective effects of *Ginkgo biloba* phytosomes (Ginkoselect Phytosome^®^, Indena) were investigated in rats with isoproterenol-induced cardiotoxicity (85 mg/kg). The oral treatment with phytosomes (100 mg and 200 mg/kg) for 21 days lowered the serum levels of AST, LDH and creatine phosphokinase (CPK) as markers of myocardial injury. Moreover, the levels of lipid peroxides decreased and those of GSH, SOD, CAT, GPx and GR increased in the myocardium of treated rats compared to untreated animals [216]. In addition to the hypoglycemic, hypolipidemic, immunomodulatory and hepatoprotective effects of *Ocimum sanctum* (Tulsi), good cardioprotective activity due to its antioxidant activity was reported [217,218,219]. The constituents of *O. sanctum*, such as flavonoids (orientin, vicenin), phenolic compounds (eugenol, cirsilineol, apigenin) and anthocyanins, have been shown to scavenge lipid peroxides [220]. The cardioprotective activity of Ginkoselect Phytosome^®^ supplemented with *O. sanctum* leaf extract was tested in isoproterenol-induced myocardial necrosis in rats. The results show that the co-administration of phytosomes (100 mg/kg) and extract (50 and 75 mg/kg) for 30 days to rats with myocardial necrosis induces the decrease in serum enzymatic markers of stressed myocardium (AST, LDH and CPK) and reduced the level of myocardial MDA, as a lipid peroxidation marker. A significant restoration of isoproterenol-affected activities and levels of AST, LDH, CPK, GSH, SOD, CAT, GPx and GR in the hearts of treated rats was measured, the most effective combination being 100 mg/kg phytosomes plus 75 mg/kg extract. The combined treatment failed to enhance the cardioprotective activity of either herbs when used individually [221]. Thus, it was demonstrated that phytosome formulation and combination with *O. sanctum* extract augmented the cardioprotective and antioxidant properties of individual ingredients.

*Withania somnifera*, known as Ashwagandha, has medicinal potential as an anti-inflammatory, anti-platelet, antihypertensive, hypoglycemic and hypolipidemic agent. Phytosomal complexes of *W. somnifera* root extract have been developed as NMITLI-101, NMITLI-118 and NMITLI-128 products (CSIR, New Delhi, India). To test the neuro-protective potential against experimental stroke, a study was conducted on a middle cerebral artery occlusion (MCAO) model in rats. The phytosomal complex NMITLI118RT+ was orally administrated at the dose of 85 mg/kg, 1 h prior to ischemia and 6 h post reperfusion. The results evidence a reduction in the neurological deficit score (over 60%) and decrease in the infarct size (over 50%), along with a 35% reduction in MDA levels and over 50% increased of GSH level [222]. 

*Carthamus tinctorius*, known as safflower, has been widely used for the treatment of cardio-cerebrovascular diseases, particularly cerebral infarction or cerebral ischemia-reperfusion (CIR) injury [223,224]. The effect of safflower extract–phospholipid complexes delivered via nasal route was tested on a rat model of MCAO and compared to the effect of unformulated extract. The therapeutic effect of phytosomes was more effective on cerebral infarction and acts by improving the blood circulation in the central nervous system, reducing the inflammatory reaction and inhibiting apoptosis. The mechanism responsible for these beneficial effects involves the caspase 3-mediated TNF-α/MAPK signaling pathway [68]. This study provides additional proof that the formulation of plant extracts in phytosomes increases their therapeutic potential.

Silybin is obtained from silymarin and is used as hepatoprotective agent [225], but its water solubility is poor and requires frequent administration [226]. The mechanisms by which silybin–phosphatidylcholine phytosomes reduce liver damage include the diminution in oxidative stress, lipid peroxidation and collagen accumulation in animal models [227]. The neuroprotective action of the silybin phytosomes (20 mg/kg per oral) against CIR injury was investigated in Wistar rats by the group of Pasala P.K. et al. [228]. The CIR injury was induced after 14 days of silybin pre-treatment by occlusion of bilateral common carotid arteries for 30 min followed by 4 h of reperfusion. Phytosomes treatment significantly increases SOD and GSH levels, and decreases MDA, TNFα and IL-6 levels in the hippocampus and cortex of treated CIR-injured rats. Histopathological studies confirmed the beneficial effects of phytosomes treatment by the alleviation of cortex cell death. In silico studies for proteins (TNFα, IL-6) involved in cerebral ischemia revealed that phytosomes exhibit strong binding interaction with them compared to thalidomide (positive control). Silybin formulation in phytosome increases its bioavailability and improves its beneficial actions compared to treatment with silybin alone [228]. These data confirm the enhancement of the therapeutic properties of bioactive compounds through their formulation in phytosomes.

The phytosomes containing extracts of ginger rhizome and mulberry fruit have been shown to decrease oxidative stress and inflammation [229]. The neuroprotective effect against ischemic stroke of phytosomes with ginger and mulberry fruit extracts was studied in male Wistar rats with metabolic syndrome induced by high-carbohydrate high-fat diet. Various doses of phytosomes (50–200 mg/kg) have been administered for 21 days before and 14 days after occlusion of the right middle cerebral artery. The results evidence that phytosomes treatment significantly enhanced memory, decreased acetylcholinesterase (AChE), IL-6 and MDA levels, and increased SOD, CAT and GPx levels, neuron density and phosphorylation of extracellular signal-regulated kinase (ERK). These data suggest the cognitive enhancing effect of phytosomes. The possible underlying mechanisms may occur partly via the improvement in cholinergic function via the ERK pathway, together with the decrease in neurodegeneration induced by the reduction in oxidative stress and inflammation [230]. In another study, phytosomes with ginger and mulberry fruit extracts significantly improved brain infarction, brain edema and neurological deficit score. In addition, the reduction in DNA-methyltransferase 1 (DNMT-1), NF-κB, TNFα and CRP, together with the increased activity of the aforementioned antioxidant enzymes and PPARγ expression in the lesioned brain, were also measured. The possible underlying mechanism may occur partly via the suppression of DNMT-1, giving rise to the improvement in the signal transduction via PPARγ, resulting in the decrease in inflammation and oxidative stress [231].

L-Carnosine and Aloe vera are food supplements used to enhance the exercise performance of athletes and to manage DM based on their potential to scavenge free radical and to exert anti-inflammatory activities [232,233,234,235]. It was demonstrated that nanophytosomes with L-carnosine/Aloe vera (25:1 *w*/*w* ratio) protect human umbilical vein endothelial cells (HUVECs) against methylglyoxal-induced toxicity following 24–72 h exposure. Nanophytosome-treated HUVECs (500 μg/mL) exhibited a greater ability in free radical scavenging, NO synthesis, tube formation, wound healing and trans-well migration compared to cells treated with unformulated constituents. The proangiogenic effects of the nanophytosomes have been confirmed by the enhanced gene expression of the following proangiogenic proteins: hypoxia-inducible factor 1-alpha (HIF-1α), vascular endothelial growth factor A (VEGF-A), basic fibroblast growth factor (bFGF), vascular endothelial growth factor (VEGF) receptor 2 (KDR) and angiotensin II (Ang II) [236]. These data empower the dual L-carnosine/Aloe vera nanophytosomes as potential therapy to attenuate the T2DM-associated microvascular complications with a reduced angiogenesis.

The *Rauvolfia serpentina* root extract contains bioactive compounds, such as flavonoids, N-containing indole alkaloids and phenols, and is effective in the treatment of diseases connected with the central nervous system, such as hypertension, insomnia and epilepsy [237]. The constituents such as reserpine, ajamlicine, serpentine, ajmaline deserpidine and yohimbine alkaloids are responsible for its anti-hypertensive properties [238,239]. The nanoformulation of the *R. serpentina* crude extract shows a lower potential to block the angiotensin-converting enzyme compared to the unformulated extract (73.99% versus 83.11%) at 5 mg/mL concentration, but was non-hemolytic compared to the extract that shows 4.31% hemolysis, as was demonstrated on human erythrocytes [240]. Thus, the formulation in phytosomes made the *R. serpentine* extract safer, despite the 10% diminution in its efficacy.

#### 4.6.2. Clinical Studies

Coenzyme Q10 (CoQ10) is able to prevent the injuries induced by free radicals and inflammatory processes, but it is known as having low bioavailability [241,242]. The formulation of coenzyme Q10 into phytosomes increased its bioavailability by three times versus standard pharmaceutical formulations and opened new opportunities for using it in clinical practice [242]. The effects of acute and chronic supplementation with CoQ10 phytosome on the endothelial reactivity and total antioxidant capacity were evaluated in a clinical study conducted in 20 healthy young non-smoking subjects. They received 150 mg CoQ10 phytosome (equivalent to 30 mg CoQ10; Ubiqsome^®^, Indena SpA) or placebo pills, once daily for the chronic group, and a double dose for the acute group. CoQ10 phytosome improved endothelial reactivity, mean arterial pressure and total antioxidant capacity compared to baseline or placebo [243]. Further clinical studies are needed to strengthen the therapeutic potential of CoQ10 formulated into phytosomes and to expand the pathologies that can be addressed with this treatment.

The confirmed beneficial effects of bioactive compounds formulated in phytosomes to improve the MD outcomes are schematically summarized in Figure 4. 

The preclinical and clinical studies that evaluated the therapeutic potential of phytosomes and their mechanisms of action are summarized in Table 4.

## 5. Final Discussion

The phytosome formulation can save and amplify the therapeutic potential of phytochemicals, being a very promising delivery system that ensures the entrapment capacity and biocompatibility in safe conditions. In the present review, we documented and highlighted the studies which demonstrate the improved therapeutic qualities of the bioactive compounds formulated in phytosomes. They can be used to counteract risk factors that induce metabolic disorders, or to treat as complementary therapy the already installed diseases. Many products based on phytosome formulation of bioactive compounds have been prepared in the last two decades and proved the in vitro and in vivo enhanced bioavailability of the carried phytochemicals, which consequently manifested the enhanced therapeutic effects of the phytoconstituents. 

The most investigated bioactive compounds formulated in phytosomes for their beneficial effects in the treatment of MD are curcumin, silymarin, ginkgo biloba extract, berberine and ginger extract. Their mechanisms of action involve the upregulation of antioxidant enzymes SOD, CAT and the GSH system, the increase in anti-inflammatory Nrf2 and NRLP3, the decrease in the pro-inflammatory cytokines, the induction of M2 macrophages, lipid-lowering effects, increase in glucose uptake, reduction in hepatic transaminases and regeneration of the liver tissue, all of which are associated with the decrease of visceral fat, improvement in the hemodynamic parameters, angiogenesis and significant recovery of infarcted tissues (Figure 4).

Remarkably, almost all reviewed studies demonstrate a good tolerability of the phytosome formulation. Many of them show significantly improved effects compared to the unformulated compounds or extracts, especially for those addressed to oxidative and inflammatory stress, hepatic disorders, dyslipidemia and CVD. Great care must be taken in future studies to establish the correct administered doses and the concentration of active principles when comparing the effects of compounds formulated in phytosomes with those unformulated. It is very important to verify the possible harmful effect of a high ratio of phospholipids, which may become a potential source of lipid peroxides. 

The results obtained so far in human subjects are promising. It is mandatory to extend such studies on larger cohorts of subjects or patients or for longer periods of time. Thus, additional clinical trials are necessary, but with improved and standardized study design, with very well characterized formulations of the phytosomes in terms of concentration of the active compounds and deliverable doses. The encouraging results obtained following the use of phytosomes should stimulate scientists to further develop improved formulations of delivery for natural bioactive compounds to ensure their increased bioavailability and improve compliance of patients at risk for MD. 

## 6. Conclusions

Many bioactive compounds or plant extracts are known to exert multiple health beneficial actions, having antioxidant, anti-inflammatory, lipid- or glucose-regulatory properties, but their low solubility and stability restrict the application of these compounds in therapy. In the present review, we emphasize the evidence supporting the use of the formulation of bioactive compounds in phytosomes with the aim of increasing their therapeutic potential. The encouraging results obtained in preclinical and clinical studies following the use of phytosomes with bioactive compounds should stimulate scientists to further discover and evaluate such products, and physicians to design larger trials in order to test them. This will strengthen the confidence of physicians and patients in the improved therapeutic effects of these phyto-phospholipid complexes. It is safe to say that treatment with bioactive compounds formulated in phytosomes should be encouraged as complementary therapy in MD patients, as well as in all subjects at risk.

## Figures and Tables

**Figure 1 ijms-25-04162-f001:**
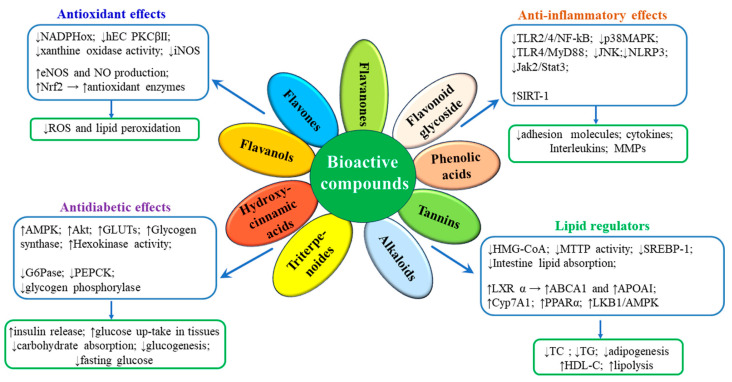
Schematic representation of the signaling pathways by which phytochemicals exert their therapeutic actions in MD. ABCA1, ATP-binding cassette A1; AMPK, AMP-activated protein kinase; ApoA1, Apolipoprotein A-I; CYP7A1, cholesterol 7alpha-hydroxylase; eNOS, endothelial nitric oxide synthase; ERK, extracellular signal-regulated kinase; GLUTs, glucose transporters; G6Pase, Glucose-6-phosphatase; HDL-C, high-density lipoproteins cholesterol; hECs, human endothelial cells; HMG-CoA, hydroxymethylglutaryl coenzyme A; iNOS, inducible nitric oxide synthase; Jak2/Stat3, Janus kinase 2/Signal transducer and activator of transcription 3; LKB1, Liver kinase B1; JNK, c-Jun N-terminal kinases; LXRs, liver X receptors; MAPK, mitogen-activated protein kinases; MD, metabolic disorders; NADPHox, nicotinamide adenine dinucleotide phosphate (NADPH) oxidase; MMPs, Matrix metalloproteinases; MTTP, Microsomal triglyceride transfer protein; MyD88, NF-κB, nuclear factor kappa B; NLRP3, nucleotide-binding oligomerization domain (NOD)-like receptor protein 3; NO, nitric oxide; Nrf2, nuclear factor erythroid 2-related factor 2; p38MAPK, p38 mitogen-activated protein kinases; PEPCK, phosphoenolpyruvate carboxykinase; PKB/Akt, Protein kinase B; PKCβII, protein kinase CβII; PPARα, peroxisome proliferator-activated receptor α; ROS, reactive oxygen species; SIRT1, sirtuin 1; SREBP, Sterol regulatory element-binding protein; TC, total cholesterol; TGs, triglycerides; TLR2/4, Toll-like receptor 2/4; upwards arrow (↑) represents increases; downwards arrow (↓) represents decreases.

**Figure 2 ijms-25-04162-f002:**
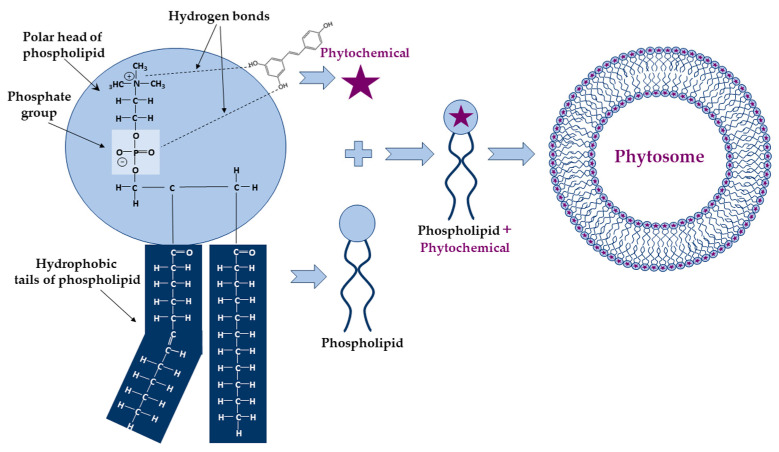
Schematic representation of the chemical bonds established during phytosome formation and their structure. The phytochemical (e.g., resveratrol) and the polar head of the phospholipid form hydrogen bonds which are represented by dashed lines.

**Figure 3 ijms-25-04162-f003:**
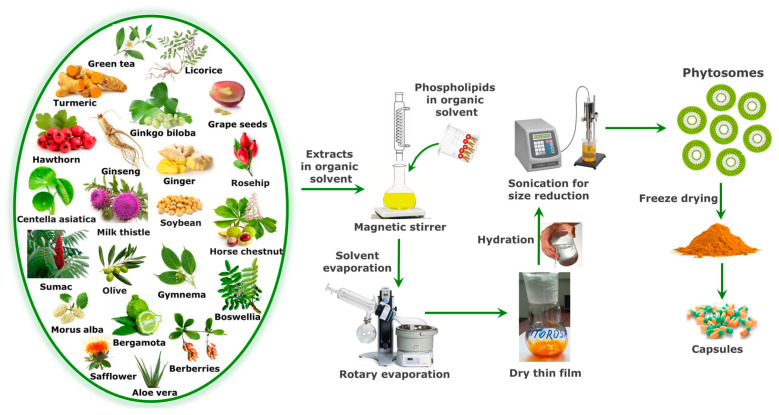
The main steps for phytosome preparation by solvent evaporation/thin-film method.

**Figure 4 ijms-25-04162-f004:**
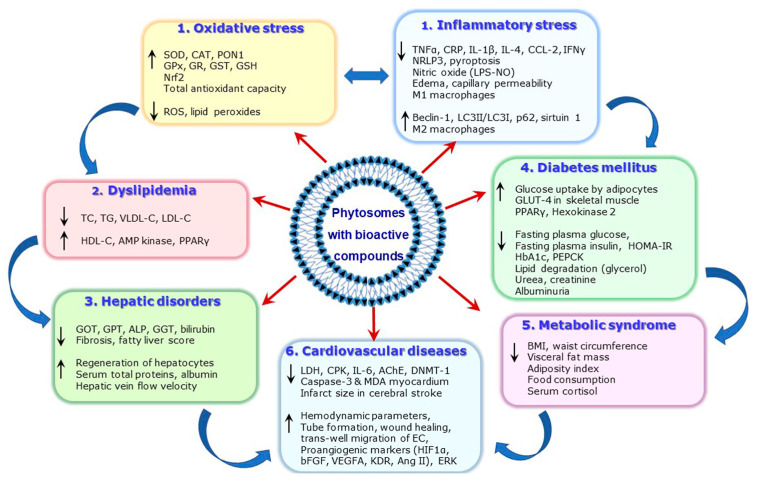
Validated beneficial effects and action mechanisms of bioactive compounds formulated in phytosomes to improve the metabolic disorder outcomes; upwards arrow (↑) represents increases; downwards arrow (↓) represents decreases.

**Table 1 ijms-25-04162-t001:** General methods for the characterization of phytosomes containing bioactive compounds.

Parameter	Techniques	References
Average size and shape	Dynamic light scattering (DLS) Scanning electron microscopy (SEM) Transmission electron microscopy (TEM) Cryo-TEM and freeze-fracture-TEM Fluorescence microscopy Atomic force microscopy (AFM) Field flow fractionation Size-exclusion chromatography	[18,38,45,46,47] [18,48,49,51,52] [18,47,48,49,50,52] [59] [52,60] [53] [54] [54]
Surface charge	DLS	[51,52,55,56]
Chemical composition and structure	Gas chromatography-mass spectrometry (GC-MS) High-performance liquid chromatography (HPLC) Fourier-transform infrared spectroscopy (FTIR) ^1^H NMR (Nuclear Magnetic Resonance spectroscopy) ^13^C NMR (Nuclear Magnetic Resonance spectroscopy) ^31^P NMR (Nuclear Magnetic Resonance spectroscopy) Differential scanning calorimetry (DSC) Powder X-ray diffraction (PXRD)	[61] [38,46,52] [47,49,51,52,61,62] [63,64,65] [65] [38] [47,49,51,52,62] [47,49,52]
Stability	^31^P (Nuclear Magnetic Resonance spectroscopy) Electron microscopy methods DSC DLS UV–Vis	[38] [48,59] [62] [46,55,56] [18,47,58]
Encapsulation efficiency (EE%) and release behavior	Mini-column centrifugation HPLC UV–Vis Dialysis Enzymatic assays Gel electrophoresis Liquid chromatography mass spectrometry	[18,38,52,57,58] [38,46,49,52] [58] [38,46,49,52] [18] [18] [51]

**Table 2 ijms-25-04162-t002:** Parameters to be measured in order to establish the level of bioavailability for various bioactive compounds formulated in phytosomes.

Subject	Dose of Bioactive Compound	Cmax ± SD Unformulated Bioactive Compound	AUC ± SD Unformulated Bioactive Compound	Tmax ± SD (h) Unformulated Bioactive Compound	Cmax ± SD Bioactive Compound in Phytosome	AUC ± SD Bioactive Compound in Phytosome	Tmax ± SD Bioactive Compound in Phytosome	References
Sprague Dawley rats	Curcumin 340 mg/kg	6.5 ± 4.5 (nM)	4.8 (µg * min/mL)	30 (min)	33.4 ± 7.1 (nM)	26.7 (µg * min/mL)	15 (min)	[70]
Rats	Berberine 50 mg/kg	66.01 ± 15.03 (ng/mL)	384.45 ± 108.62 (ng * h/mL)	0.5 (h)	219.67 ± 6.02 (ng/mL)	1169.19 ± 93.75 (ng * h/mL)	2 (h)	[52]
Rats	Apigenin 100 mg/kg	0.14 ± 0.15 (µg/mL)	0.84 ± 0.42 (μg * h/mL)	2.0 ± 0.23 (h)	0.20 ± 0.25 (µg/mL)	1.31 ± 0.46 (μg * h/mL)	4.0 ± 0.34 (h)	[43]
Sprague Dawley rats	Baicalein 75 mg/kg	1.61 ± 0.37 (µg/mL)	664.68 ± 75.50 (µg * min/mL)	170.00 ± 65.73 (min)	8.68 ± 1.35 (µg/mL)	1748.20 ± 280.80 (µg * min/mL)	33.33 ± 8.67 (min)	[71]
Rats	Ursolic acid 20 mg/kg	8 ± 0.21 (µg/mL)	13.15 ± 0.34 (μg * h/mL)	1.5 ± 1.23 (h)	9 ± 2.17 (µg/mL)	60.33 ± 2.19 (μg * h/mL)	2 ± 0.43 (h)	[72]
Male Albino rats	Resveratrol 25 mg	0.24 ± 0.12 (µg/mL)	24.31 ± 4.31 (µg * min/mL)	30 (min)	2.27 ± 0.51 (µg/mL)	257.15 ± 40.26 (μg * min/mL)	60 (min)	[73]
Human	Curcuminoids 376 mg	5.2 ± 0.2 (ng/mL)	39.6 ± 1.5 (ng * h/mL)	9.5 ± 0.2 (h)	8.7 ± 0.4 (ng/mL)	65.3 ± 2.3 (ng * h/mL)	1.7 ± 0.4 (h)	[74]
Human	Curcuminoids 376 mg	0.9 ± 0.1 (ng/mL)	10.4 ± 1.3 (ng * h/mL)	4 (h)	18.0 ± 6.4 (ng/mL)	86.9 ± 12.1 (ng * h/mL)	1 (h)	[67]
Human	Coenzyme Q10 30 mg	0.10 ± 0.05 (µg/mL)	1.43 ± 1. 96 (μg * h/mL)	16.83 ± 19.73 (h)	0.13 ± 0.08 (µg/mL)	3.92 ± 3.56 (µg * h/mL)	24 ± 18.68 (h)	[75]
Human	Quercetin 500 mg	10.93 ± 2.22 (ng/mL)	4774.93 ± 1190.61 (ng * min/mL)	290 ± 31.19 (min)	223.10 ± 16.32 (ng/mL)	96,163.87 ± 9291.31 (ng * min/mL)	202.50 ± 35.97 (min)	[76]
Human	Curcumin 207 mg	2.03 ± 1.79 (nM)	19.06 ± 17.47 (nM * h)	6.92 ± 5.96 (h)	16.61 ± 10.10 (nM)	147.9 ± 67.84 (nM * h)	6.92 ± 8.34 (h)	[77]
Human	Berberine 188 mg	69.95 ± 14.54 (pg/mL)	1057 ± 117 (pg * h/mL)	4.55 ± 0.29 (h)	375.57 ± 41.56 (pg/mL)	4146 ± 431 (pg * h/mL)	4.50 ± 0.30 (h)	[78]

AUC; Area under the curve (0-t); C_max_; Maximum concentration of the drug in plasma; T_max_; Time of maximum drug concentration measured in the plasma.

**Table 3 ijms-25-04162-t003:** Commercial products based on phytochemicals formulated in phytosomes and their beneficial effects in MDs.

No.	Commercial Product; Doses and Administration Protocol	Company	Phytochemicals	Beneficial Effects; Mechanisms of Action	References
1	Berberine–phospholipid complex-based phytosomes (100 mg/kg orally administered to db/db mice 4 weeks)	-	Berberine	Anti-diabetic (decreased glucose levels in plasma and TG in the liver)	[52]
2	Casperome^®^ phytosome (250 mg/day orally administered to subjects with musculoskeletal conditions, 1–4 weeks)	Indena	*Boswellia serrata*–Resin	Anti-inflammatory (lower the pain score, decrease CRP levels)	[79]
3	Curcumin phytosome (1 g × 2/day, corresponding to 200 mg curcumin, orally to humans with acute muscle injury, 4 days)	Indena	*Curcuma longa* L.-Rhizome	Anti-inflammatory (decreased IL-8)	[80]
4	Greenselect^®^/Green Tea Phytosome (2 × 150 mg/day Greenselect and 30 mg piperine orally administered to obese women, 3 months)	Indena	*Camellia sinensis* L.-Leaf	Bodyweight regulator (reduction in weight and fat mass, improvement of lipidic profile growth hormone, insulin-like growth factor-1, insulin and cortisol)	[81,82]
5	Monoselect Camellia (MonCam) 1 tablet eq to 300 mg/day of Greenselect phytosomes for 24 weeks	PharmExtracta (Pontenure, Italy)	*Camellia sinensis*	Improver of lipidic profile, antioxidant (reduced fasting glucose, increased HDL, decreased TG, reduced plasma free radicals)	[83]
6	Hawthorn Phytosome (100 mg, orally)	Indena	Flavonoids of *Crataegus species*	Blood pressure regulator, cardioprotective	[84,85]
7	Leucoselect^®^/Grape Seed Phytosome (300 mg grape procyanidin extracts eq, to healthy smoking adults, 4 weeks)	Indena	Vitis vinifera L.–Seed	Antioxidant, cardioprotective (reduced lipid peroxidation in LDL, increased the lag phase of LDL oxidation)	[86]
8.	Ginseng Phytosome (100 or 200 mg per day for eight weeks orally administered to diabetic patients)	Natural Factors (Monroe, WA, USA)	*Panax Ginseng*	Antidiabetic effects (decreased fasting blood glucose, reduced hemoglobin A_1C_ values for 200 mg dose)	[18,87]
9	Naringenin Phytosome (Approx 3 mg Naringenin eq, intratracheary administered to rats with acute lung injury, 4 h)	-	*Citrus aurantium*	Anti-inflammatory, antioxidant (increased SOD2 mRNA, decreased COX2 and ICAM-1 gene expression, decreased p-p38MAPK in the lungs)	[88]
10	Quercefit™ Phytosome (1–2 tabs/day administered to human subjects with both asthma and rhinitis, for 30 days)	Indena	Quercetin	Antioxidant (reduction of plasma-free radicals)	[89]
11	Siliphos^®^ (120 mg, orally administered)	Indena	Silybin of *Silybum marianum*	Antioxidant, hepatoprotective	[85,90]
12	Green Tea Phytosome 1 capsule 2–3 times/day orally administered to obese subjects for 90 days)	Natural Factors	Green tea polyphenols	Antioxidant, bodyweight regulator (thermogenic effect, improvements in weight and body mass index)	[91]
13	Ubiqsome™ Phytosome (150–300 mg corresponding to 30–60 mg CoQ10, administered orally to healthy humans for two periods of two weeks)	Indena	Co-enzyme Q10	Antioxidant; a good absorbance of CoQ10	[75]
14	Vazguard™ Phytosome (1000 mg/L of bergamot phytosome subjected to simulated gastric digestion and further incubated to fecal slurries from healthy women)	Indena	Bergamot extract	Regulator of plasma glucose and lipid levels, bodyweight regulator (modulator of gut microbiota: Firmicutes, Proteobacteria, Bacteroidetes, Actinobacteria)	[92]
15	Vazguard™/Naringin Phytosome (oral administration to type 2 diabetic patients)	Indena	Bergamot extract	Antioxidant, cardioprotective (improvement in lipid profile, reduction in small, dense LDLs and decreased glucose levels in plasma)	[93]

**Table 4 ijms-25-04162-t004:** Preclinical and clinical studies evidencing the therapeutic potential of phytosomes with bioactive compounds for MD treatment.

Study Type	Disorder	Bioactive Compounds Formulated in Phytosomes; Beneficial Effects
**Preclinical**	**Oxidative and Inflammatory stress**	- ***Ginkgo biloba*** extract on Wistar rats with rifampicin-induced hepatotoxicity: ↓ plasma lipid peroxides; ↑ GSH, SOD, CAT, GPx, GR [96]; - **Curcumin** on mice with carrageenan-induced inflammation: ↑ SOD, CAT, GPx, GR [99]; - ***Zingiber officinale*** and ***Rosa canina*** extracts on C57Bl/6J mice with LPS-induced inflammation: ↓ gut and liver TNFα, IL-1β; ↑ plasma SOD2, PON1 [38] - **Gallic acid** on rats with valproic acid-induced oxidative stress: ↑ GPx, GR, GSH, SOD, CAT (Nrf2) [113] - **Berberine** on mice with inflammation: ↓ edema, local inflammation, capillary permeability [29] - ***Allium ampeloprasum*** extract on BALB/c mice with colon carcinoma: ↑ liver GPx and SOD mRNA; ↓ MDA [120] - ***Gymnema inodorum*** leaf extract on LPS-stimulated RAW 264.7 macrophages: ↓ NO [26] - **Tripterine + selenium** on J774A.1 macrophage: ↓NRLP3 inflammasome, pyroptosis, gasdermin D, IL-1β [124]
	**Dyslipidemia**	- ***Murraya koenigii*** extract on STZ-induced diabetic Wistar rats: ↓ TC, TG, LDL-C, VLDL-C [51] - ***Crataegus aronia*** leaf extract on STZ-induced diabetic Wistar rats: ↓ TC, TG, LDL-C; ↑ HDL-C [50] - **Caffeic acid** on Sprague Dawley rats fed high-fat diet: ↓ TC, TG, LDL-C, VLDL-C; ↑ HDL-C [151] - **Curcumin** on New Zealand white rabbits fed high-fat diet: ↓ atheroma, intima/media thickness, macrophage infiltration [152] - **Silymarin** on hyper-triglyceridemic rats: ↓ TC, TG; ↑HDL-C, CYP4A, CYP7A1, ABCG5/8 [153]
	**Hepatic disorders**	- ***Ginkgo biloba*** extract on Wistar rats with rifampicin/CCl_4_-induced hepatotoxicity: ↓ plasma ALT, AST, ALP, liver TBARS; ↑ serum total proteins, albumin, GSH, SOD, CAT, GPx, GR [96,160]; - ***Silybum marianum*** extract/**Silybin/Silibinin** on Wistar rats with CCl_4_-induced hepatotoxicity or BALB/c mice with D-GalN/LPS: ↓ ALT, AST, MDA, S100A9, M1 macrophages; ↑ liver SOD CAT, GPx, GST, GR, GSH, M2 macrophages [47,164,165] - **Naringenin** on Wistar rats with CCl_4_-induced hepatotoxicity: ↓ plasma ALT, AST, bilirubin; ↑ hepatic GPx, SOD, CAT [166] - **Catechin** on Wistar rats with cadmium-induced hepatotoxicity: ↓ plasma AST, ALT, bilirubin; ↑ SOD, CAT, GPx [169] - **Apigenin** on Wistar rats with CCl_4_-induced hepatotoxicity: ↓ TBARS; ↑ GSH, SOD, CAT [43] - **Ursolic acid** on Wistar rats with CCl_4_-induced hepatotoxicity: ↓ serum AST, ALT, ALP, bilirubin; ↑ liver SOD, CAT, GPx, GST, GR, GSH, total protein [72]
	**Diabetes mellitus**	- ***Murraya koenigii*** extract on STZ-induced diabetic Wistar rats: ↓ serum glucose, urea, and creatinine [51] - ***Crategus aronia*** leaf extract on STZ-induced diabetic Wistar rats: ↓ serum glucose [50] - ***Citrullus colocynthis*, *Momordica balsamina, Momordica dioica*** extract on STZ-induced diabetic Wistar rats: ↓ serum glucose [176] - **Chrysin** on db/db mice: ↓ serum glucose, insulin, HOMA-IR, hepatic PEPCK; ↑ GLUT4 [179] - **Rutin** on STZ-induced diabetic Wistar rats: ↓ serum glucose, HbA1C, TC, TG, AST, ALT, MDA; ↑ HDL-C, SOD, GPx [182] - **Curcumin** on STZ-induced diabetic Wistar rats: ↓ serum glucose, TC, TG, LDL-C; ↑ HDL-C [185] - **Berberine** on db/db mice: ↓ serum glucose, TG [52] - ***Gymnema inodorum*** leaf extract on LPS-treated 3T3-L1 adipocytes: ↑ glucose uptake [26] - **Tripterine (selenized)** on high glucose-exposed podocytes: ↑ viability; ↓ NLRP3, apoptosis [191]
	**Metabolic syndrome**	- ***Callistemon citrinus*** leaf extract on obese Wistar rats: ↓ body weight, adiposity index, TG [200] - **Soybean (*Glycine max*)** on Wistar rats with high-fat diet: ↓ body weight, adipose tissue weight, food consumption, plasma TC, TG, LDL-C, VLDL-C [69]
	**Cardiovascular diseases**	- **Gymnemic acid** on rat with cardiomyopathy doxorubicin-induced: ↓ serum LDH, myocardial caspase-3 and TBARS; ↑ Na+/K+ ATPase, SOD, CAT, GPx; improve hemodynamic parameters and heart weight/body weight ratio [213] - ***Ginkgo biloba*** (±***Ocimum sanctum***) on rats with isoproterenol-induced cardiotoxicity: ↓ AST, LDH, CPK; ↑ myocardial GSH, SOD, CAT, GPx, GR [216] - ***Withania somnifera*** root extract on MCAO rats: ↓ neurological deficit score, infarct size, MDA; ↑ GSH [222] - ***Carthamus tinctorius*** extract on MCAO rats: ↑ blood circulation in CNS; ↓ inflammation, apoptosis (caspase 3-TNF-α/MAPK) [68] - **Silybin** on CIR Wistar rats: ↑ SOD, GSH; ↓ MDA, TNFα, IL-6 in cortex and hippocampus [228] - ***Zingiber officinale*** rhizomes ***+ Morus*** fruits extracts on Wistar with metabolic syndrome: ↑ memory, SOD, CAT, GPx, neuron density, pERK; ↓ AChE, IL-6, MDA, [217]; ↑ neurological deficit score, PPARγ, ↓ DNMT-1, NF-κB, TNFα, CRP [231] - **L-Carnosine + *Aloe vera*** on methylglyoxal -HUVECs: ↑free radical scavenging, NO synthesis, tube formation, wound healing, trans-well migration, HIF-1α, VEGF-A, bFGF, VEGF receptor 2 (KDR), Ang II [236] - ***Rauvolfia serpentina*** root extract on human erythrocytes: ↓ hemolysis [237]
**Clinical**	**Oxidative and Inflammatory stress**	- **Grape seeds** extract on young healthy volunteers: ↑serum total antioxidant activity [132] - **Curcumin** on chronic kidney disease patients: ↓ plasma CCL-2, IFNγ, IL-4, TBARS, gut *Escherichia-Shigella;* ↑ gut *Lachnoclostridium*, *Lactobacillaceae* spp. [140]
	**Dyslipidemia**	- ***Citrus bergamia*** on overweight and obese class I subjects: ↓ TC, LDL-C; ↑ HDL-C [156]
	**Hepatic disorders**	- **Curcumin (±piperine)** on patients with NAFLD: ↓ fibrosis, liver parenchyma echogenicity, fatty liver score, AST, ALT, GGT, TG, BMI; ↑ HDL-C [172,173,174]
	**Diabetes mellitus**	- **Quercetin** on diabetic patients: does not interfere with antidiabetic or anticoagulant therapies [192] - **Curcumin** on kidney dysfunction, overweight or NAFLD subjects: ↓ plasma free radicals; ↑kidney function [195]; ↓ serum glucose, insulin, HOMA-IR, cortisol [174]; ↓ AST, NAFLD grade [196] - **Berberine** on polycystic ovary syndrome patients: ↓ serum glucose, insulin, HOMA-IR, VLDL-C, TG, TNFα, CRP, visceral fat [187]
	**Metabolic syndrome**	- **Curcumin** on subjects with metabolic syndrome: ↑ antioxidant effects independent on vitamin E [206] - ***Citrus bergamia*** on overweight and obese class I subjects: ↓ visceral adipose tissue [156] - ***Camellia sinensis*** extract on subjects with metabolic syndrome: ↓ body weight, waist circumference, plasma free radicals [83]
	**Cardiovascular diseases**	- **Coenzyme Q10** on healthy young subjects: ↑ endothelial reactivity, arterial pressure, total antioxidant capacity [243]

## Data Availability

Not applicable.

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
