# Peer review of "Bioactive Compounds Formulated in Phytosomes Administered as Complementary Therapy for Metabolic Disorders"

_ijms, 2024, doi:10.3390/ijms25084162_

Round 1

Reviewer 1 Report

Comments and Suggestions for Authors

The current manuscript is an interesting review on phytosomes containing bioactive compounds for the treatment of metabolic disorders. It appears to be overall well-structured and written. Hence, only some alterations are necessary before acceptance for publication:

- In the introduction section, a schematic image should be produced and added regarding the action mechanisms of the mentioned phytocompounds in the treatment of metabolic diseases;

- In the introduction section, more should be said about nanosystems for therapeutic purposes, including their general characteristics (for example size and uses), advantages and disadvantages, enumeration of the different types of nanosystems that exist (inorganic and organic, and then divided into nanoparticles, micelles, liposomes, etc.), and only then does it make sense to talk about phytosomes, which should also be defined as a derivative of liposomes;

- An image regarding phytosomes structure, composition and characteristics should be made and added in the introduction section;

- In section 2.1., more should be said about phytosomes preparation methods (deeper description);

- In section 3, more should be said regarding administration routes;

- Why are the only commercial products that are mentioned from the same brand (Indena)? Examples from different brands should be added;

- In general Figure quality (resolution) should be improved, maybe by making them bigger;

- The “Conclusion” section is really a “Final discussion” section, so it should be renamed; additionally, a much shorter and summarized “Conclusion” section should be made and added afterwards;

- An abbreviation list is missing and should be added.

Reviewer 2 Report

Comments and Suggestions for Authors

Comments for Authors

This article entitled “Bioactive compounds formulated in phytosomes administered as complementary therapy for metabolic disorders” is interesting by its subjects. In this review, authors explain how phytochemicals in phytosomal formulations can be administered in combination with allopathic drugs as complementary remedies for metabolic disorders.

The authors provide valuable insights and commentary regarding prior work on intentionality, helping to enhance its significance and utility. However, the article needs extensive editing before proper acceptance. The authors are advised to follow the following recommendations to improve the quality of the manuscript;

Abstract

The abstract should be more concise. Remove typos and grammatical errors. For example, Line 13, alkaloids, triterpenes should be changed with alkaloids, and triterpenes

Line 17, “or large chemical structure” change with “and complex chemical structure”

Keywords: Use keywords derived from international information network searches to increase visibility. It is more transparent if you use alphabetical order of some of the used keywords.

Introduction

Line 38-40, rephrase the following sentence “Others, although successful, induce …” for clarity.

Please kindly clearly specify what is the rationale of the study

Was the literature background systematic? Kindly provide search strategies.

Line 109, “In Figure 1 are presented” remove “are”

Line 147, IR spectra? The abbreviation should be added in parentheses after the written-out form. Please check and revise the entire manuscript.

There are too many descriptions on Characterization of phytosomes with bioactive compounds and their information is not the newest. The authors should focus on “Effect of phytosome in bioactive compounds”, but not “Introduction of Characterization of phytosomes with bioactive compounds. Different studies should be added to explore the characterization of phytosomes with different bioactive compounds.

Table 1, Phytosomes with names of bioactive compounds should be added in 1st column.

Table 3, There is no highlight about the mechanism in the table, more details such as the mode of administration and dose should be listed.

Line 260, cite the citation number [24] as per journal style “Deleanu et al. [24]”

Line 268, The scientific name of sumac (Rhus coriaria L.) could be added in text.

 Line 288-290, The statement “This is why the absolute bioavailability of berberine is no more….” should be rephrased for better understanding.

Line 469, the The scientific name of Bergamot (Citrus bergamia) should be added.

Line no 682-688: Reframe the sentence.

The introduction section needs further improvement, such as a more comprehensive discussion on the pathological mechanisms of Oxidative and inflammatory stress, Dyslipidemia, Hepatic disorders, Diabetes mellitus and Cardiovascular disorders.

The shortcomings and countermeasures of existing researches should be discussed, and the future research development could be out looked.

There should be a new table added before the conclusion, which summarizes preclinical and clinical studies, health perspectives, and basic mechanisms of action of phytosomes bioactive compounds.

Conclusion: Line 1155-1157, Reframe the statement “It is safe to say that Treatment with bioactive compounds formulated in phytosomes should be encouraged as complementary therapy in MD patients, as well as in subjects at risk.

The English syntax should be check by a native English speaker.

Comments on the Quality of English Language

The article needs extensive editing of English language before proper acceptance. The English syntax should be check by a native English speaker.

Round 2

Reviewer 2 Report

Comments and Suggestions for Authors

The authors have addressed all raised concerns. Therefore, the manuscript can be accepted.